# Geochemical Signature and Magnetic Fabric of Capinha Massif (Fundão, Central Portugal): Genesis, Emplacement and Relation with W–Sn Mineralizations

**Ana Gonçalves** [1,2,*] **, Helena Sant'Ovaia** [1,2] **and Fernando Noronha** [1,2]

[1] Departamento de Geociências, Ambiente e Ordenamento de Território, Faculdade de Ciências da Universidade do Porto, Rua do Campo Alegre 687, 4169-007 Porto, Portugal; hsantov@fc.up.pt (H.S.); fmnoronh@fc.up.pt (F.N.)

[2] Instituto de Ciências da Terra (ICT), Pólo da Universidade do Porto, Rua do Campo Alegre 687, 4169-007 Porto, Portugal

\* Correspondence: ana.goncalves@fc.up.pt

**Abstract:** The Fundão–Serra da Estrela–Capinha (FSEC) region is characterized by peraluminous to metaluminous Variscan granites intrusive in a complex and thick metasedimentary sequence. This work seeks to characterize the Capinha granite (CG), understand its spatial and genetic relationship with the host Peroviseu–Seia (PS), Belmonte–Covilhã (BC) and Fáguas granites, and evaluate its metallogenic potential. To achieve these goals, a multidisciplinary approach was undertaken, including field work and identification of the petrography and microstructures, whole rock geochemistry and anisotropy of magnetic susceptibility. Four distinct and independent differentiation trends were identified in the granites, namely, PS, BC, Fráguas and CG. The PS and BC played a role as host rocks for the W and Sn mineralizations. The Fráguas granite is anomalous in Sn and spatially related to the Sn–Li mineralizations, while the CG is anomalous in W and spatially related to W–Sn mineralizations. The post-tectonic CG is a peraluminous ilmenite-type whose ascent and emplacement were tectonically controlled. The Capinha magma used the intersection between the 25° N and 155° N strike–slip crustal scale faults for passive ascent and emplacement during the late-Variscan extensional phases. The magnetic fabric was drawn using an asymmetric tongue-shaped laccolith for CG. CG experienced two brittle deformation stages that marked the maximum compressive rotation from NE–SW to NNW–SSE.

**Keywords:** Variscan granites; differentiation trends; W and Sn mineralizations; post-tectonic Capinha granite; passive emplacement; tongue-shape

## 1. Introduction

The production and emplacement of granitic magmas are well represented in orogenic belts, playing an important role in continental recycling and crustal growth. Granites are good markers of crustal kinematics and allow one to reconstruct the tectonic evolution of orogenic belts, e.g., [1–3].

Granites also act as a source of ore metals and mineralized fluids and/or as a source of heat that generates the large convective fluid circulation responsible for ore deposit formation. Granite's geochemical signatures give important information about (i) the composition of initial granite magmas, (ii) the involved differentiation processes and (iii) the degree of evolution related to the concentration of Sn and W in residual evolved melts and/or the fluids developed during fractional crystallization, e.g., [4–6]. The reconstruction of the evolution of granite intrusions is grounded in

the interpretation of the internal fabric patterns and microstructures developed during the successive stages of ascent—emplacement and post-emplacement together with the study of the geochemical evolution of granite intrusion. Fabric patterns can be precisely studied using the anisotropy of magnetic susceptibility (AMS) technique [7,8]. The use of AMS studies is crucial for granitic rocks since, at a macroscopic scale, the preferred orientation of iron-bearing minerals is weakly marked or absent [8–10]. Thus, granite's kinematic may be inferred by mapping its internal magnetic fabric and anisotropy magnitudes, which can reflect the magmatic strain pattern of the magma [11–15].

The magnetic susceptibility ($K_m$) obtained from AMS measurements has an important role in the distinction between "magnetite-type" (ferromagnetic behavior, [10]) and "ilmenite-type" (paramagnetic behavior, [10,12]) granites [16]. Several studies have been done to constrain the limits between para- and ferromagnetic behaviors. The authors in [17] suggested that paramagnetic behavior is below 1000 µS.I., the authors in [18] suggested that the limit is near 300 µS.I., the authors in [19] proposed a boundary close to 200 µS.I., and the researchers in [20] found paramagnetic behaviors in leucogranites bellow 100 µS.I. This range of values is directly related to the contents of ilmenite and magnetite in the accessory fraction of the granite composition. Thus, the "magnetite-type" contains about 0.1 vol%–2.0 vol% of magnetite, and the "ilmenite-type" presents only ilmenite at less than 0.1 vol% [17]. According to the work in [17] and [21], "magnetite-type" magma may be generated at low depths (lower crust to upper mantle), where no carbonaceous material exists. Magnetite- and ilmenite-type granite classification is important to understand the redox conditions in the magma, as well as to infer what kind of mineral occurrence is associated with each type of granite [17].

In the Iberian Variscan Belt, the processes responsible for the genesis of compositionally distinct granitic magmas involves the partial melting of distinct rock sources, hybridization, and/or the mixing of contrasting magmas, fractional crystallization, crustal contamination and assimilation-fractional crystallization [22–32].

The NW area of the Iberian Peninsula is characterized by the existence of a W–Sn metallogenic province associated with granite intrusions [33–35]. The W–Sn deposits occur in greisens and/or quartz veins related to late-orogenic biotite-rich granites and biotite-muscovite granites [36–38].

The Fundão–Serra da Estrela–Capinha region is extremely important from the perspective of W–Sn ore deposits, as it contains not only the largest underground W–Sn mine in the Iberian Peninsula (the Panasqueira mine), but also several old mines and small occurrences of W–Sn that were intensively explored in the first half of the last century.

In this article, we study a small-scale intrusion whose emplacement was structurally controlled and spatially associated with W–Sn mineralizations—the Capinha granite located in the Fundão–Serra da Estrela–Capinha region. The fieldwork, petrography, microstructures, whole rock geochemistry, and AMS data of the Capinha granite used to achieve these main goals are reported in this study. The Capinha granite offers a good opportunity to better understand the structural constraints of the emplacement of post-tectonic Variscan granite under extensional settings, determine its spatial and genetic relationship with the surrounding r its metallogenic potential.

## 2. Regional Geology

The Iberian Variscan Belt (IVB) (or Iberian Massif) is a well-exposed section of continental crust resulting from a collision of the Gondwana and Laurussia supercontinents. This continental collision halted the closure of the Rheic Ocean during the Late Devonian to Middle Carboniferous, followed by a post-thickening extension from the Middle Carboniferous to the Permian [39–43].

Three main regional ductile deformation phases have been described ($D_1$, $D_2$ and $D_3$) [40,41,44–46] in NW Spain and N and Central Portugal, corresponding to the Galiza-Trás-os-Montes Zone (GTMZ) and the Central Iberian Zone (CIZ) [47,48]. $D_1$ and $D_2$ were responsible for the greatest amount Variscan crustal growth. $D_3$ is correlated with the post-thickening and extension that marked the end of continental collisions [48,49]. $D_3$ produced vertical folds with a sub-horizontal axis and subvertical dextral and sinistral strike–slip shear zones [50], which were responsible for the emplacement of

a large volume of orogenic granites. The ages of $D_1$, $D_2$ and $D_3$ are 360–337 Ma, 337–316 Ma and 315–310 Ma, respectively [32,51]. The authors in [52], based on the Tourmaline $^{40}Ar/^{39}Ar$ chronology of tourmaline-rich rocks, dated the main Variscan deformation phases as $D_1$—370 ± 5 Ma and $D_2$—342 ± 5 Ma, and obtained an age of ca. 296 Ma for tourmalines in the youngest leucogranites. The emplacement of lamprophyre dykes at ca. 265 Ma [53] reveals an extension-related crustal melting setting typical of the late Variscan stages.

In the allochthonous domain of IVB (i.e., in the GTMZ), three deformational phases are evidenced, where $D_2$ is associated with thrusts. In the allochthonous domain (i.e., in the CIZ), only the $D_1$ and $D_3$ deformation phases are represented [43]. $D_3$ is followed by brittle deformation phases post-$D_3$ (310–270 Ma, [54]), which are responsible for a set of conjugate strike–slip faults (NNW–SSE, NNE–SSW and ENE–WSW) related to the maximum N–S shortening [46,54–57].

Based on fieldwork observations and geochronological data, the granites situated in the CIZ and GTMZ were divided into six main groups according to their age of emplacement relative to $D_3$ [58–61]: pre-$D_3$ (331–321 Ma); syn-$D_3$ (321–312 Ma); late-$D_3$ (312–305 Ma), late- to post-$D_3$ (ca. 300 Ma); and post-$D_3$ (<299 Ma), where granites younger than 310 Ma are considered post-collisional (Figure 1).

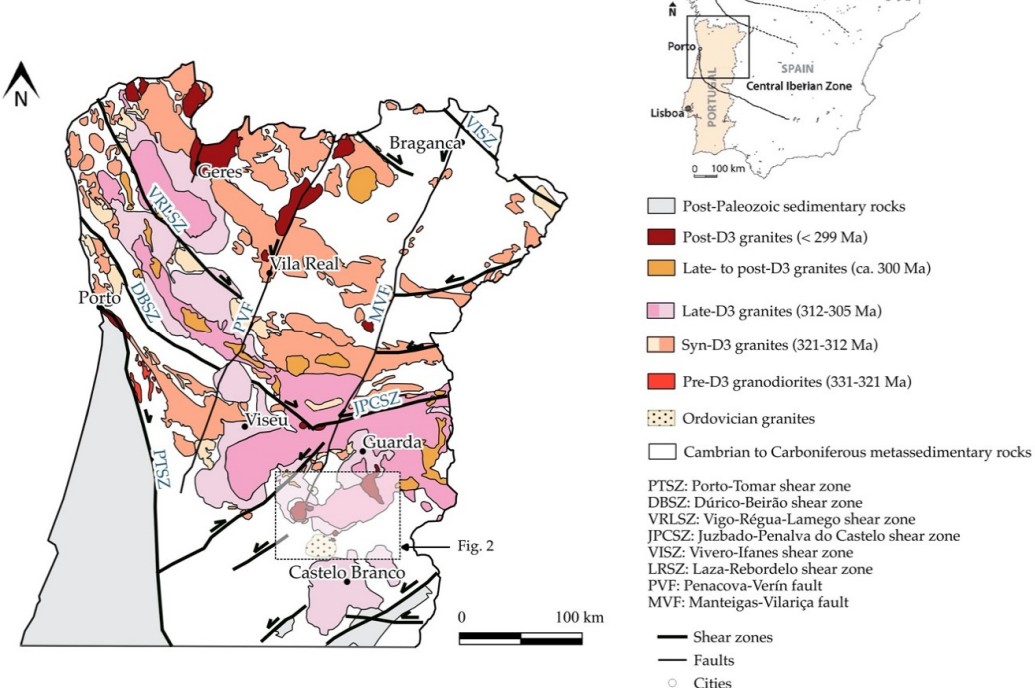

**Figure 1.** Spatial distribution of the Variscan granites and the major structures in central and northern Portugal (adapted from [58]).

## 3. Fundão–Serra da Estrela–Capinha Region

### 3.1. Geology and Field Relationships

The Fundão–Serra da Estrela–Capinha region (FSEC) is located in the CIZ, Central Portugal. This region is characterized by the occurrence of several outcrops belonging to a set of different intrusive granitic massifs in metasedimentary sequences from the Beiras group (the "Schist Greywacke Complex" (SGC), which is Neoproterozoic in age) [62–65]. The SGC sequences displays dominant NW–SE trending foliation ($S_1$) with vertical or subvertical dips (80° to 85° SW). The study area is cut by an important NNE–SSW regional fault, the Vilariça–Manteigas fault (VMF).

The intrusive granites from FSEC correspond to four different groups: Ordovician granites, late-$D_3$, late- to post-$D_3$ and post-$D_3$ granites (Figure 2).



- The Ordovician group is represented by Manteigas granite (482 ± 6 Ma, [27]), which is a biotite-rich, coarse-grained, porphyritic granodiorite with clear foliation with tonalites and monzonitic granites that constitute the composite and zoned pluton of Fundão [66–68]. Recent geochronological data yielded an Ordovician age (478.1 ± 0.5 Ma and 478.4 ± 0.5 Ma, respectively, for tonalite and monzogranite) for the Fundão pluton [69].

- The late-D$_3$ group is represented by Peroviseu (on the east side of VMF) and Seia (on the west side of VMF), which comprise the Peroviseu–Seia granite (PS) and Castelo Branco granite (CB). The PS is biotite-rich, coarse- to very coarse-grained porphyritic granite. The distribution of K feldspar is generally random, although an E–W magma flow, marked by slightly oriented K feldspar, was recognized. These granites occasionally have some enclaves with granodioritic to tonalitic composition. CB is a complex zoned pluton; from the core to the periphery, it is composed of muscovite–biotite granite, biotite ± muscovite porphyritic granodiorite and muscovite–biotite granite.

- The late- to post-D$_3$ group is represented by biotite-rich monzonitic granites (Belmonte–Covilhã granite). Belmonte–Covilhã (BC) is a biotite-rich, coarse-grained, porphyritic monzogranite and displays well-defined deformation patterns with oriented K feldspar megacrysts. The contacts between the BC and PS are, generally, gradual. On the other hand, the contact between the BC and the metasediments have an NE–SW trend and are characterized by the occurrence of mottled schists and hornfels due to the thermal metamorphism related to the BC emplacement.

- The post-D$_3$ granite group is composed of two-mica leucogranites—Capinha (CG), Estrela, Fráguas, Atalaia and non-outcropping Panasqueira granite.

  ○ CG is a muscovite > biotite, medium-grained, incipient porphyritic granite. CG occurs as a small circular circumscribed body exposed over an area of about 7 km$^2$, intruding into the low-grade metamorphic Schist–Greywackes of SGC, ~15 km to the north–east of Fundão village (Figure 2). CG intrudes the contact between the PS granite and the SGC and cuts the foliations of the SGC. In the whole extension, the contacts with steeply dipping metasediments are sharp and nearly vertical, with no apparent structural deformations, suggesting that the granite was not forcibly emplaced. The textural characteristics of CG are homogeneous in the whole area, without any magmatic flow foliation or superimposed deformation.

  ○ The two-mica leucogranite group is characterized by irregular bodies intruding in the BC, Fráguas (on the east side of VMF), Estrela (on the west side of VMF) and Atalaia (on the east side of VMF). The Estrela muscovite-rich granite is located at the highest point of the Portugal mainland (~2000 m high) in the Serra da Estrela mountain. Fráguas granite is irregular and elongated NW–SE, suggesting structural control of its emplacement. The dominant facies are composed by two-mica, medium-grained granites, although coarse-grained facies also occur. Fráguas granite exhibits a gradual petrographic zonation corresponding to the biotite-rich facies in the central part and muscovite-rich facies in the peripheries [70]. The Estrela and Fráguas granites occupy the dome areas of the BC biotite-rich porphyritic granite. Atalaia granite is a tourmaline–muscovite, medium-grained, porphyritic granite located in the north–eastern side of the Fundão pluton [66,68].

  ○ In the Panasqueira area, where no granites outcrop, underground mining crosses a greisen cupula. Several authors [71–79] suggested that this greisen cupula is connected to a non-outcropping granite dome. Subsequent research surveys intercepted a two-mica fresh granite (with muscovite that is dominantly secondary in origin), which was dated in [80] to ca. 289 ± 4 Ma (whole-rock Rb–Sr) with a high $^{87}Sr/^{86}Sr_i$ ratio of 0.713.

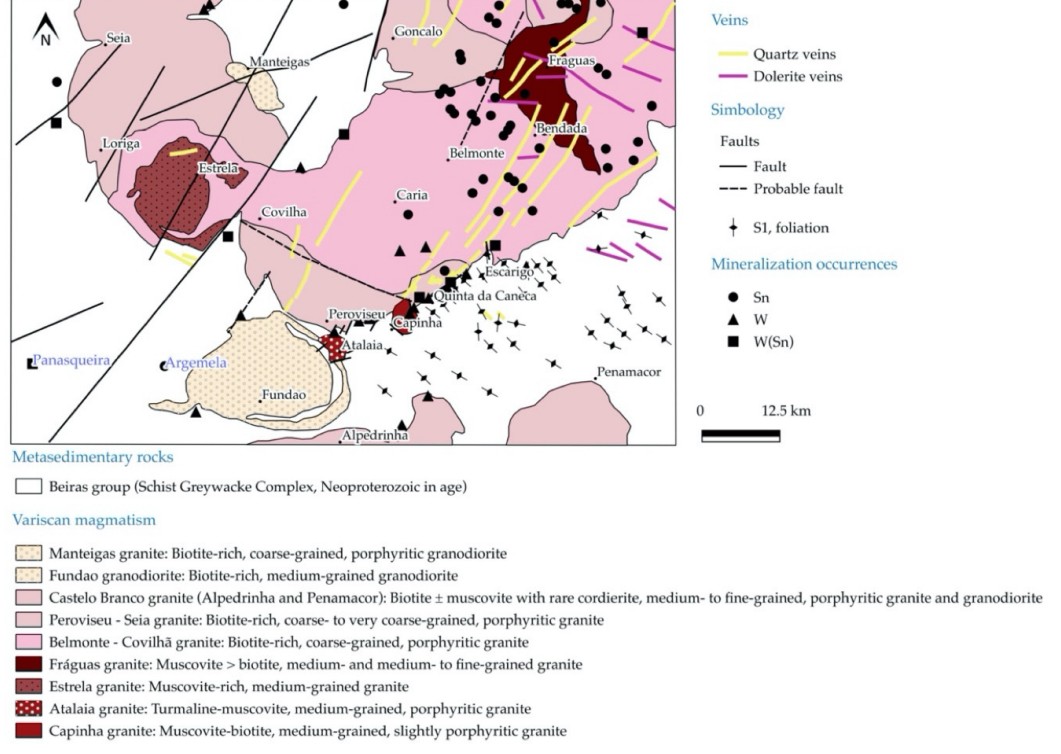

**Figure 2.** Simplified Fundão–Serra da Estrela–Capinha region geological map and Sn, W and W(Sn) occurrences associated with Variscan granites [58,67,82].

In Table 1 is described the general features of the outcropping lithologies in the Fundão-Serra da Estrela-Capinha region.

**Table 1.** General features of the outcropping lithologies in the Fundão–Serra da Estrela–Capinha region [81,82]. n.d.—not determined.

| Lithology | Dating | Description | Mineralogy |
|---|---|---|---|
| Schist–Greywacke Complex (SGC) and derivate metamorphic series | | | |
| Schists and greywackes | Neoproterozoic | Sequences characterized by the interlayering of fine-to-medium-grained, gray to greenish color greywackes and fine- to medium-grained shining phyllites. | The main mineralogy of greywackes is quartz, K feldspar, chlorite and sericite; this biotite can occur in zones where the metamorphic effect is more intense. The schist is mostly composed of quartz, biotite and muscovite; K feldspar, tourmaline and graphite are present as accessories in mineralogy. |
| Ordovician granites | | | |
| Manteigas granite | 482 ± 6 Ma [27] | Biotite-rich, coarse-grained, porphyritic granodiorite with clear foliation. | Quartz, microcline, plagioclase, biotite, chlorite, sphene, apatite, zircon, rutile, magnetite and ilmenite. |
| Fundão granite | 478.1 ± 0.5 Ma and 478.4 ± 0.5 Ma for tonalite and monzogranite, respectively [69] | The Fundão intrusive complex consists of granodioritic zoned pluton. The central part features granodiorites and an outermost zone of biotite granite. | Quartz, K feldspar (microcline), plagioclase (oligoclase-andesine), biotite, muscovite, apatite, topaz, allanite, epidote, sphene, rutile, ilmenite and pyrite. |
| Late-D$_3$ granites | | | |
| Peroviseu–Seia granite (PS) | 304.1 ± 3.9 Ma [83] | Biotite-rich, coarse- to very coarse-grained porphyritic granite. | Quartz, microperthite microcline, plagioclase, biotite, chlorite, muscovite, zircon, apatite, monazite, ilmenite, rutile and rare cordierite. |
| Castelo Branco granite (CB) (Alpedrinha and Penamacor) | 310 ± 1 Ma [84] | CB is a complex zoned pluton composed by muscovite–biotite granite in the core, which is surrounded by biotite ± muscovite porphyritic granodiorites and muscovite–biotite granite in the peripheries. | Quartz, microperthitic microcline, plagioclase, biotite, some chlorite, muscovite, tourmaline, monazite, apatite, zircon, ilmenite and rutile. The CB also contains magmatic andalusite, sillimanite and cordierite. |
| Late- to post-D$_3$ granites | | | |
| Belmonte–Covilhã granite (BC) | 300 ± 1 Ma [28] | Biotite-rich, coarse-grained, porphyritic granite. | Quartz, K feldspar, plagioclase, biotite ± muscovite ± chlorite, zircon, rutile, sphene, tourmaline, ilmenite, fluorite and opaques. Rare andalusite can occur. |
| Post-D$_3$ granites | | | |
| Fráguas | 299 ± 3 Ma [28] | Muscovite > biotite, medium- and medium- to fine-grained granites. | Quartz, K feldspar (microcline, albite, albite–oligoclase and microperthite), fresh biotite, zircon. The K feldspars are strongly altered to sericite and kaolin. |
| Atalaia granite | n.d. | Tourmaline muscovite, medium-grained, porphyritic granite. | Quartz, K feldspar, albite, muscovite or bleached biotite and tourmaline. |
| Estrela granite | n.d. | Muscovite-rich, medium-grained granite. | Quartz, K feldspar (microcline), albite muscovite and tourmaline. |
| Capinha granite | n.d. | Muscovite > biotite, medium-grained, incipient porphyritic granite. | Quartz, K feldspar (microcline >> orthoclase), plagioclase (albite–oligoclase), micas (muscovite > biotite ± chlorite), apatite, zircon, rutile and metallic phases. The plagioclase is strongly altered to sericite and kaolin. |

*3.2. Regional Brittle Deformation*

The Fundão area includes four main subvertical fracturing systems (60° N–70° E, 100° N–110° E, 30° N–40° E and 150° N–160° E) and also sub-horizontal joints (Figure 2). These fracturing systems affect the whole area with distinct intensities, conditioning different geological processes, as follows:

1. The 60° N–70° E fracturing system is well marked in the contact between PS and BC biotite-rich porphyritic granites and metasedimentary rocks.
2. The 100° N–110° E fractures control the emplacement of the PS granites recorded in the E–W magmatic flow by the K feldspar orientation and its reactivation during the late-Variscan stages, which promoted the emplacement of Sn aplite-pegmatites and W mineralized quartz veins.
3. The 30° N–40° E and its conjugate 150° N–160° E are well represented in the study area by the regional quartz veins with post-Variscan U mineralizations.

*3.3. Mineralizations*

The "Iberian Sn–W metallogenic province" [33], included in the "Northern mineralogenic province" [34], is located in the Variscan Iberian belt and is characterized by the occurrence of several W and Sn ore deposits. The Sn and W primary ore deposits are related to magmatic–hydrothermal systems commonly associated with granites. The magmatic differentiation processes promote the concentration of W and Sn in the residual evolved melts, which later precipitate in the apical granitic zones. In N and Central Portugal, Sn mineralizations are mainly associated with peraluminous two-micas granites and W mineralizations associated with biotite rich granites ([36,37] and the references therein).

The FSEC region, part of the Iberian Sn–W metallogenic province, is characterized by several Sn and W occurrences (Figure 2). The spatial distribution of the distinct types of mineralizations is not accidental. The Panasqueira mine, which is currently active, is one of the most important W–Sn ore deposits in the world. Its mineralization is associated with a greisen cupola related to the depth of the granitic body. Mineralization occurs in sub-horizontal quartz veins that fill a network of sub-horizontal joints in the metasedimentary sequences of SGC.

## 4. Materials and Methods

*4.1. Sampling Procedure*

Sampling for the petrographic and Anisotropy of Magnetic Susceptibility (AMS) analyses was carried out using the Capinha granite; 30 sites were sampled with a total of 227 oriented cores (Figure 3). For the whole rock geochemistry, two samples were collected from an active quarry (Q1 and Q2). The geographic distribution of the sampling was conditioned according to the difficult accessibility of outcrops and the weathering degree. The geographic coordinates of each sampling site are presented in Table A1 and Figure A1. Nevertheless, an attempt was made to homogenize sampling across the CG (about ~1 sampling site/0.23 km$^2$). At each site, 4 or 5 cylinders 25 mm in diameter were drilled using a Stihl petrol-powered portable drill and oriented in situ using a magnetic compass corrected to the local magnetic declination (~1°47′ W).

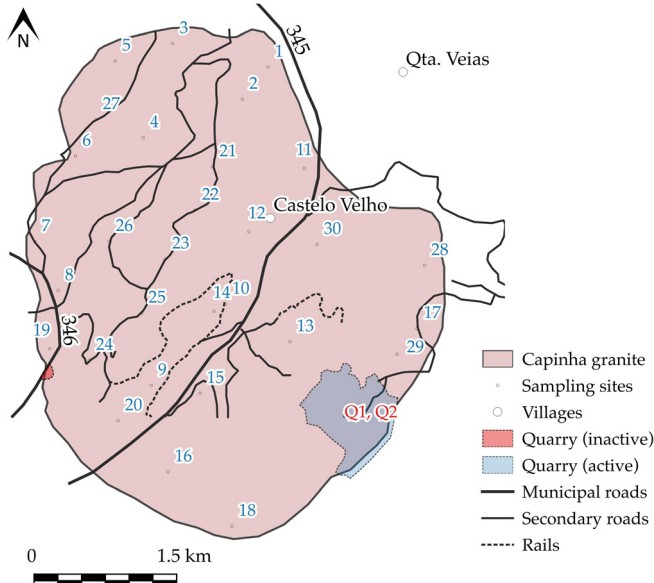

**Figure 3.** Location of the sampling sites for petrographic, anisotropy of magnetic susceptibility and whole rock geochemistry studies in Capinha granite.

### 4.2. Petrography and Microstructures

For petrographic, microstructural and microfracture studies, thin sections were taken from selected oriented cores. Petrography was then performed using a Zeiss Stemi SV11 Apo stereomicroscope coupled with a Sony Cyber-Shot DSC-S75 digital camera using a Leica DM LSP polarizing microscope with transmitted and reflected light, coupled with a Leica camera with the LAS EZ software 2.0.0 at the Department of Geosciences, Environment and Spatial Planning, and Institute of Earth Sciences (Faculty of Sciences, University of Porto). The study of opaque mineralogy was performed with a scanning electron microscope/energy dispersive x-ray spectroscope (SEM-EDS) at the Department of Geosciences, Environment and Spatial Planning, and Institute of Earth Sciences (Faculty of Sciences, University of Porto) using a FLEXSEM1000. The SEM-EDS was operated at 10.00 kV in the low-vacuum mode, with a manual aperture and 4.5 beam spot sizes. The abbreviations used to identify the different mineral phases followed the nomenclature proposed in [85].

### 4.3. Whole Rock Geochemistry

The whole rock chemical compositions of the 8 crushed samples from CG (95% < 74 μm) were analyzed for their major ($SiO_2$, $TiO_2$, $Al_2O_3$, $Fe_2O_3$, FeO, MnO, MgO, CaO, $Na_2O$, $K_2O$ and $P_2O_5$), trace (Nb, Zn, Sn, Zr, Y, Sr, Ba, Rb, Th and W), and rare-earth elements (La, Ce, Nd, Sm, Eu, Gd, Tb, Dy, Ho, Er, Tm, Yb and Lu) at Activation Laboratories Lda., Ontario, Canada. The samples were fused using lithium metaborate ($LiBO_2$) and digested using a nitric acid solution ($HNO_3$). Solutions were analyzed by inducing coupled plasma optical emission spectrometry (ICP-OES) for major elements, ICP mass spectrometry (ICP-MS) for trace- and rare earth elements and titration (TITR) for FeO. The loss on ignition (LOI) was determined gravimetrically using a precision scale. The uncertainties of major elements were generally between 1% and 3%, whereas most of the trace elements had uncertainties below 10%. More information about the ICP-OES, ICP-MS and TITR analytical procedures, precision values and limits of detection is given by the Activation Laboratories Lda. official page [86].

### 4.4. Anisotropy of Magnetic Susceptibility

AMS measurements were performed on 160 oriented specimens using a KLY-4S Kappabridge susceptometer ($\pm 3.8 \times 10^{-4}$ T; AGICO Ltd., Brno, Czech Republic) at the Department of Geosciences, Environment and Spatial Planning and the Institute of Earth Sciences (Faculty of Sciences, University

of Porto) operating at room temperature with a 300 A/m AC field at 920 Hz. Using a sequence of 15 susceptibility measurements along different orientations for each specimen allowed us to compute the orientation and magnitude of the three main axes ($K_1 \geq K_2 \geq K_3$) of the AMS ellipsoid by applying Jelinek statistics [87]. Tensor reconstruction and statistical analyses were made using the ANISOFT 4.2 software [88]. AMS defines the magnetic susceptibility tensor that correlates the strength of the applied field, $H$ (A/m), to the acquired magnetization, $M$ (A/m), of a material, expressed by the mathematic equation:

$$M = KH \tag{1}$$

Thus, the $K$ or $K_m$ (dimensionless in SI units) is a 2nd-rank tensor originating from a symmetric matrix that describes the spatial variation of magnetic susceptibility (e.g., [8]). This tensor is expressed by the magnitude (eigenvalues) and orientation (eigenvectors) of the maximum ($K_1$), intermediate ($K_2$) and minimum ($K_3$) principal axes of the magnetic ellipsoid. The mean magnetic susceptibility is calculated according to the follow equation:

$$K_m = \frac{K_1 + K_2 + K_3}{3}. \tag{2}$$

An AMS ellipsoid is a geometrical representation of such a tensor, where $K_1$ represents magnetic lineation and $K_3$ is the pole of magnetic foliation (e.g., [8,20]). The shape of the magnetic susceptibility ellipsoid is represented by the shape parameter ($T$) from [84], expressed by

$$T = \frac{ln(F) - ln(L)}{ln(F) + ln(L)} \tag{3}$$

$$F = ln\left(\frac{K_2}{K_3}\right) \tag{3a}$$

$$T = ln\left(\frac{K_1}{K_2}\right) \tag{3b}$$

where $F$ is planar anisotropy and $L$ is linear anisotropy, which can be $T = +1$ (pure planar or oblate ellipsoid), $T = -1$ (pure linear or prolate ellipsoid) or $T = 0$ (triaxial ellipsoid). The magnitude of the anisotropy degree ($P$) corresponds to $K_1/K_3$. This study uses the parameter $P_{para}$:

$$P_{para}\% = 100 \times \left[\left(\frac{K_1 - K_d}{K_3 - K_d}\right) - 1\right] \tag{4}$$

where $K_d$ corresponds to the diamagnetic components of quartz and feldspar [18]. This parameter, $P_{para}$, is more realistic than $P$ for rocks with paramagnetic behavior, thus making it necessary to subtract $K_d$, which is constant ($-14.6 \times 10^{-5}$ SI) and isotropic, to avoid artificial enhancement of paramagnetic anisotropy [89].

## 4.5. Fracturing

The samples used in microfracturing studies have neither strong superficial alterations nor (generally) hand fractures. The strikes of the microfractures (or fluid inclusion planes, FIP) were measured on each oriented thin section using the PLANIF program [90,91]. For the microfracturing study, we considered intergranular and intercrystalline microfractures [92] healed by secondary fluid inclusions [93,94] without evidence of lateral movement. Statistical interpretation of the data was made using rose diagrams, in which the microcracks are given in 10° classes using the "Stereonet 10.0.6 64-bit version" program [95–97].

## 5. Results

### 5.1. Main Petrographic and Mineralogic Features

#### 5.1.1. Petrography and Microstructural Study

The CG displays a typical hypidiomorphic inequigranular medium- to fine-grained texture. The petrography allowed us to identify the presence of quartz, alkali feldspar (microcline >> orthoclase) and plagioclase (albite–oligoclase), comprising 36.45%, 26% and 26.64% of the total normative composition, respectively (Table 2). Muscovite, biotite, chlorite, apatite, rutile, zircon, monazite and opaque minerals are ubiquitous accessory minerals. A complete petrographic and microstructural description of the studied samples is provided in Table 3 and illustrative microphotographs are presented in Figure 4.

**Table 2.** Normative Cross–Iddings–Pirsson–Washington (CIPW) composition of the Capinha granite. Nomenclature [85]: Qz—quartz; Pl—plagioclase; Or—orthoclase; Crn—corundum; Hyp—hypersthene; Ilm—ilmenite; Mag—magnetite; Hem—hematite; Ap—apatite. Min—minimum; Max—maximum; $\bar{x}$—average; σ—standard deviation; n—number of samples.

| Mineral% | Capinha Granite ($n = 8$) | | | |
|---|---|---|---|---|
| | Min | Max | $\bar{x}$ | σ |
| Qz | 35.29 | 37.58 | 36.45 | 0.7 |
| Pl | 25.31 | 28.26 | 26.64 | 1 |
| Or | 25.18 | 26.69 | 26 | 0.52 |
| Crn | 3.93 | 5.34 | 4.68 | 0.4 |
| Hyp | 1.02 | 3.47 | 2.19 | 0.75 |
| Ilm | 0.4 | 0.66 | 0.45 | 0.08 |
| Mag | 0.49 | 1.83 | 1.1 | 0.45 |
| Hem | 0 | 1.35 | 0.17 | 0.45 |
| Ap | 0.53 | 0.88 | 0.8 | 0.11 |

**Table 3.** Mineralogy and microstructures associated with the growth and deformation of minerals in the CG. The distinguished domains were defined by considering a distinct range of temperatures, namely, (i) a magmatic to submagmatic state for felsic magmas: T > 750 °C [3,98] and (ii) a solid state at a low-temperature: T < 400 °C [99]. Nomenclature [85]: Quartz (Qz), Orthoclase (Or), Microcline (Mc), Albite (Ab), Biotite (Bt), Chlorite (Chl), Muscovite (Ms), Zircon (Zrn), Apatite (Ap), Rutile (Rt).

| Mineral | Occurrence | Magmatic to Submagmatic | Low-T Solid State | Other Observations |
|---|---|---|---|---|
| Quartz (Qz) | Anhedral to subhedral inequigranular crystals; very variable sizes ranging from a few μm up to 500 μm. | No preferred orientation; inclusions in other igneous crystals; evident undulose extinction (Figure 4a). | Scarce subgranulation and sutured boundaries with recrystallized bulges (Figure 4b); strongly fractured (Figure 4c). | Local, poorly defined chess-board patterns are present. |
| K feldspar | The orthoclase (Or) and microcline (Mc) occur in well-developed crystals; microcline is more abundant than orthoclase and exhibits crystals up to 500 μm. Orthoclase occurs in small crystals, ca. 200 μm. | Eu- to subhedral crystals; growth twins common in both K feldspars; orthoclase displays perthites, venules and bands (Figure 4d); eu- to subhedral inclusions in other igneous crystals, generally albite–oligoclase. | Slightly glide and/or deformed twins (Figure 4e); altered to fine-grained white micas (sericitization). | Occasionally, the orthoclase is replaced by microcline, displaying Carlsbad twins together with cross-hatched twins (Figure 4f); microcline and orthoclase exhibit a poikilitic texture characterized by small inclusions of quartz, orthoclase and micas. |
| Plagioclase (Pl) | Plagioclase, namely, albite–oligoclase, occurs in well-developed crystals (≥500 μm). | Eu- to subhedral crystals; slightly zoned (oscillatory); growth twins common parallel to the long axis; eu- to subhedral inclusions in other igneous crystals, generally, in the microcline. | Strongly altered to very fine-grained white micas (sericitization) and clays (Figure 4g); deformed twins (Figure 4h); fracturing infilled by iron oxides. | Sporadically, metal phases occur associated with the plagioclase; commonly, the replacement of the albite–oligoclase develops through the zoning planes (Figure 4i). |
| Phyllosilicates | Three types were identified: primary muscovite (Ms I, the most abundant), biotite (Bt) and chlorite (Chl) (Figure 4j). They occur in anhedral to subhedral crystals with frayed ends, isolated or in clusters of several flakes; sizes ranging between 100 and 500 μm. | Ms and Bt are, generally, subhedral, with well-defined cleavage; randomly distributed; eu- to subhedral inclusions in other igneous crystals. | Muscovite presents slight deformation (kinking) (Figure 4k); biotite is, frequently, replaced by chlorite and also by secondary muscovite (Ms II). | Inclusions of small clusters of eu- to subhedral zircons (lengths lower than 50 μm) in Bt, promoting the origin of pleochroic halos; inclusions of subhedral platy crystals of ilmenite (sizes lower than 100 μm) parallel to the cleavage planes of biotite and/or chlorite; very fine-grained reeds of rutile (lower than 30 μm in length) associated with the chloritization of Bt; inclusions of apatite (100 to 200 μm) and monazite (< 30 μm) in Bt, showing euhedral to subhedral shapes (Figure 4l). |

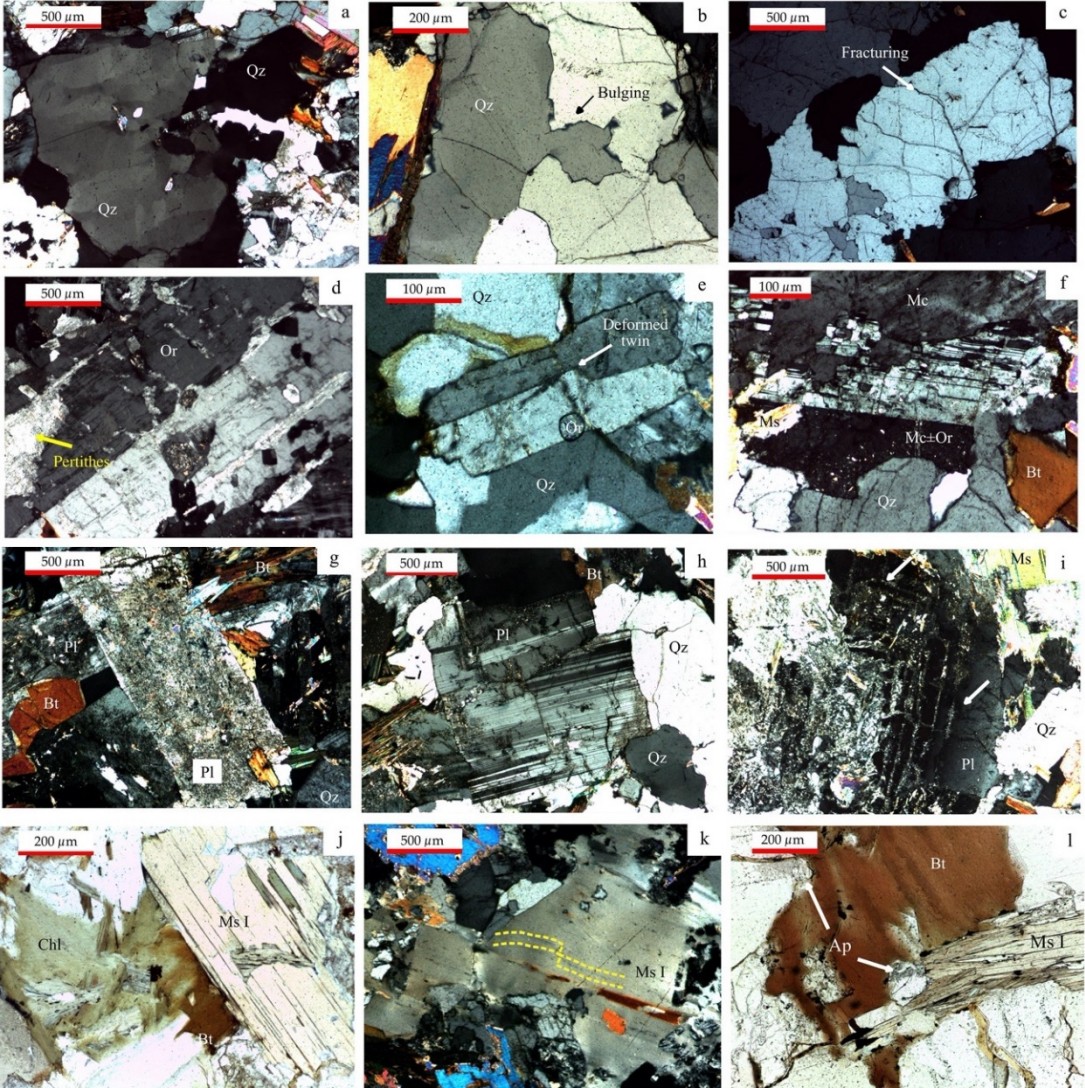

**Figure 4.** Microstructures and textural features observed in the main and accessory mineralogy. (**a**–**c**) Quartz: undulose extinction (**a**), sutured boundaries with recrystallized bulges (**b**) and strong fracturing (**c**); (**d**–**f**) K feldspar: orthoclase displaying perthites (**d**), deformed twins (**e**) and the replacement of orthoclase with microcline, displaying Carlsbad twins together with cross-hatched twins (**f**); (**g**–**i**) plagioclase strongly altered to very fine-grained white micas (sericitization) and clays (**g**), evidence of deformed twins (**h**) and alterations in albite–oligoclase developing through the zoning planes (**i**); (**j**–**l**) micas: relationships between primary muscovite (Ms I), biotite (Bt) and chlorite (Chl) (**j**); slight deformation of muscovite cleavages (**k**); and inclusions of apatite (100 to 200 μm) in Bt (**l**). (microphotographs **a**, **b**, **c**, **d**, **e**, **f**, **g**, **h**, **I** and **k**) taken under cross polarized light (CPL). (microphotographs **j** and **l**) taken under plane polarized light (PPL).

### 5.1.2. Magnetic Mineralogy

The magnetic mineralogy was investigated in CG using optical microscopy under reflected light and SEM-EDS. The petrography of the CG revealed the prevalent occurrence of paramagnetic minerals (biotite and opaque phases). The opaque mineralogy is dominated by ubiquitous ilmenite and subordinate amounts of pyrite, arsenopyrite and hematite. Ilmenite is mainly found as eu- to subhedral grains, which are generally optically homogeneous. These grains are usually associated with phyllosilicates (mainly, biotite) that have undergone alteration: (1) small (~50 μm) and elongated segregations along the basal section of the biotite (Figure 5a) and/or chloritized biotites and (2) small

(50 to 100 μm) patches that are generally associated with apatite in the matrix (Figure 5b–d). Pyrite is also observed as irregular crystals associated with apatite in connection with a widespread and usually strong, alteration of the plagioclase (~500 μm), (Figure 5e). Little arsenopyrite appears in the eu- to subhedral crystals (~100 μm) disseminated in the matrix (Figure 5f).

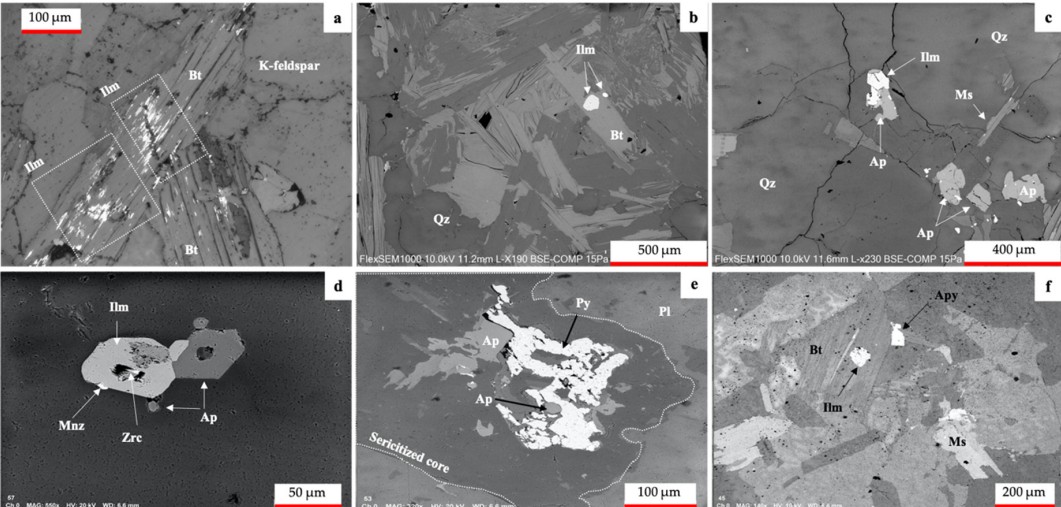

**Figure 5.** Opaque mineralogy and mineral associations. (**a**–**d**) General aspects of the distinct modes of occurrence of ilmenite (Ilm) in the small elongated crystals included in the biotite cleavages (**a**), larger single euhedral crystals (**b**) and crystals associated with apatite, zircon and monazite (**c**) and (**d**); (**e**) large anhedral crystals (>100 μm) of pyrite (Py) associated with apatite in the sericitized core of the plagioclase (albite–oligoclase); (**f**) anhedral crystals of arsenopyrite (Apy, ca. 100 μm) in the matrix. All the microphotographs were taken using a high-contrast backscattered electron (BSE) analysis, except for (**a**), which was taken under reflected light.

### 5.1.3. Capinha Granite: Geochemical Characterization

Representative data of the major, trace and rare earth elements from the Capinha granite are reported in Tables 4 and 5. Samples 1, 4, 11, 15, 18 and 19 were collected at the outcrop surface (568 m altitude) and samples Q1 and Q2 were collected from a new quarry located in the southeastern boundary, at a depth of 87 m (481 m altitude). The whole rock composition shows that the granite is strongly homogeneous in the whole outcrop surface and also in depth (Figures 6 and 7). The CG is characterized by $SiO_2$ contents ranging between 72.07 wt% and 73.41 wt% (Figure 6a,b) and $Al_2O_3$ contents from 14.33 wt% to 15.1 wt%. (Table 4). The CG is enriched in $K_2O$ (4.26 wt%–4.5 wt%) and $P_2O_5$ (0.33 wt%–0.38 wt%). On the other hand, the CG is impoverished in $Fe_2O_3$ (0.55 wt%–1.69 wt%), FeO (0.3 wt%–1.5 wt%), CaO (0.5 wt%–0.73 wt%), $NaO_2$ (2.93 wt%–3.32 wt%) and $TiO_2$ (0.21 wt%–0.23 wt%) compared to the world average granite composition (WAGC) [99,100] (Table 4). For trace elements, the CG is depleted in Sr (61–74 ppm), Y (11–13 ppm), Zr (71–79 ppm), Nb (9–10 ppm), Ba (239–263 ppm), Ta (2–2.3 ppm) and Th (6.3–7.2 ppm) (Table 4). In contrast, the CG is enriched in Rb (239–268 ppm), Sn (9–24 ppm), W (6–12 ppm) and U (10.1–15.8 ppm). The K/Rb ratio ranges from 131.96 to 149.02.

**Table 4.** Major (wt%) and trace (ppm) element data for Capinha granite. Notes: (1) when the analyzed values are below the lower limit, the values are considered half the lower limit's value; (2) the major and trace elements of the world average granite composition (WAGC) were compiled from the literature [100,101]. LOI—lost on ignition; —not determined; Min—minimum; Max—maximum; $\bar{x}$—average; σ—standard deviation.

| Element | Capinha Granite | | | | | | | | Min | Max | $\bar{x}$ | σ | WAGC |
|---|---|---|---|---|---|---|---|---|---|---|---|---|---|
| | 1 | 4 | 11 | 15 | 18 | 19 | Q1 | Q2 | | | | | |
| SiO$_2$ | 72.57 | 73.34 | 72.07 | 72.87 | 72.44 | 73.41 | 72.76 | 72.61 | 72.07 | 73.41 | 72.76 | 0.42 | 72.04 |
| Al$_2$O$_3$ | 14.82 | 14.97 | 14.33 | 15.1 | 14.97 | 14.75 | 14.94 | 14.51 | 14.33 | 15.1 | 14.8 | 0.24 | 14.42 |
| Fe$_2$O$_3$ | 0.97 | 0.66 | 1.69 | 1.04 | 1.26 | 1.03 | 0.89 | 0.55 | 0.55 | 1.69 | 1.01 | 0.33 | 1.22 |
| MnO | 0.051 | 0.034 | 0.037 | 0.031 | 0.043 | 0.036 | 0.038 | 0.038 | 0.031 | 0.051 | 0.04 | 0.01 | 0.05 |
| FeO | 1.1 | 1.3 | 0.3 | 1 | 0.8 | 1.1 | 1.2 | 1.5 | 0.3 | 1.5 | 1.04 | 0.34 | 1.68 |
| MgO | 0.5 | 0.44 | 0.41 | 0.51 | 0.44 | 0.54 | 0.67 | 0.67 | 0.41 | 0.67 | 0.52 | 0.09 | 0.71 |
| CaO | 0.54 | 0.6 | 0.6 | 0.5 | 0.62 | 0.73 | 0.61 | 0.63 | 0.5 | 0.73 | 0.6 | 0.06 | 1.82 |
| Na$_2$O | 2.96 | 3.08 | 3.22 | 3.01 | 3.32 | 3.19 | 3 | 2.93 | 2.93 | 3.32 | 3.09 | 0.13 | 3.69 |
| K$_2$O | 4.5 | 4.38 | 4.48 | 4.33 | 4.26 | 4.29 | 4.5 | 4.44 | 4.26 | 4.5 | 4.4 | 0.09 | 4.12 |
| TiO$_2$ | 0.229 | 0.231 | 0.212 | 0.223 | 0.21 | 0.223 | 0.233 | 0.231 | 0.21 | 0.23 | 0.22 | 0.01 | 0.3 |
| P$_2$O$_5$ | 0.37 | 0.38 | 0.35 | 0.33 | 0.37 | 0.36 | 0.35 | 0.36 | 0.33 | 0.38 | 0.36 | 0.01 | 0.12 |
| LOI | 1.25 | 1.32 | 1.05 | 1.26 | 0.98 | 1.02 | 1.34 | 1.56 | 0.98 | 1.56 | 1.22 | 0.18 | n.d. |
| Total | 99.86 | 100.7 | 98.76 | 100.2 | 99.72 | 100.7 | 100.5 | 100 | – | – | – | – | 100.17 |
| Sc | 5 | 5 | 5 | 5 | 5 | 5 | 5 | 5 | 5 | 5 | 5 | 0 | – |
| Be | 4 | 4 | 4 | 4 | 6 | 4 | 4 | 4 | 4 | 6 | 4.25 | 0.66 | – |
| V | 22 | 23 | 20 | 21 | 20 | 22 | 23 | 22 | 20 | 23 | 21.63 | 1.11 | – |
| Cr | 150 | 160 | 150 | 160 | 160 | 180 | 170 | 170 | 150 | 180 | 162.5 | 9.68 | – |
| Co | 5 | 4 | 3 | 2 | 7 | 4 | 4 | 4 | 2 | 7 | 4.13 | 1.36 | – |
| Ni | <20 | 60 | <20 | <20 | 20 | 30 | <20 | <20 | 20 | 60 | 36.67 | 17 | – |
| Cu | <10 | 40 | 10 | 30 | <10 | < 10 | <10 | <10 | 10 | 40 | 26.67 | 12.47 | – |
| Zn | 70 | 50 | 60 | 50 | 90 | 60 | 60 | 60 | 50 | 90 | 62.5 | 11.99 | 39 |
| Ga | 19 | 19 | 19 | 20 | 19 | 19 | 20 | 19 | 19 | 20 | 19.25 | 0.43 | – |
| Ge | 2 | 2 | 3 | 2 | 2 | 2 | 2 | 2 | 2 | 3 | 2.13 | 0.33 | – |
| As | 18 | 16 | 72 | 42 | 49 | 19 | 44 | 42 | 16 | 72 | 37.75 | 17.99 | – |
| Rb | 258 | 247 | 258 | 249 | 268 | 239 | 253 | 248 | 239 | 268 | 252.50 | 8.26 | 170 |
| Sr | 72 | 68 | 74 | 64 | 69 | 70 | 61 | 61 | 61 | 74 | 67.38 | 4.58 | 100 |
| Y | 13 | 13 | 11 | 13 | 13 | 12 | 12 | 11 | 11 | 13 | 12.25 | 0.83 | 40 |
| Zr | 76 | 79 | 78 | 77 | 71 | 77 | 77 | 77 | 71 | 79 | 76.5 | 2.24 | 175 |
| Nb | 10 | 10 | 10 | 10 | 9 | 10 | 10 | 10 | 9 | 10 | 9.88 | 0.33 | 28 |
| Mo | 2 | 3 | 2 | 2 | 3 | 3 | 2 | 3 | 2 | 3 | 2.5 | 0.5 | – |
| Ag | <0.5 | <0.5 | <0.5 | <0.5 | <0.5 | <0.5 | <0.5 | <0.5 | – | – | – | – | – |
| In | <0.2 | <0.2 | <0.2 | <0.2 | <0.2 | <0.2 | <0.2 | <0.2 | – | – | – | – | – |
| Sn | 9 | 14 | 20 | 24 | 24 | 12 | 16 | 16 | 9 | 24 | 16.88 | 5.09 | 3 |

**Table 4.** *Cont.*

| Element | Capinha Granite | | | | | | | | Min | Max | $\bar{x}$ | $\sigma$ | WAGC |
|---|---|---|---|---|---|---|---|---|---|---|---|---|---|
| | 1 | 4 | 11 | 15 | 18 | 19 | Q1 | Q2 | | | | | |
| Sb | <0.5 | <0.5 | <0.5 | <0.5 | <0.5 | <0.5 | <0.5 | <0.5 | – | – | – | – | – |
| Cs | 14.9 | 16.7 | 15.7 | 17.6 | 21.3 | 11.3 | 18.5 | 17.9 | 11.3 | 21.3 | 16.74 | 2.74 | – |
| Ba | 263 | 250 | 261 | 241 | 248 | 247 | 239 | 239 | 239 | 263 | 248.5 | 8.72 | 340 |
| Ta | 2.3 | 2.1 | 2.1 | 2.2 | 2 | 2.3 | 2.2 | 2.2 | 2 | 2.3 | 2.18 | 0.1 | 4 |
| W | 6 | 8 | 11 | 12 | 12 | 9 | 10 | 9 | 6 | 12 | 9.63 | 1.93 | 2.2 |
| Tl | 1.6 | 1.4 | 1.6 | 1.5 | 1.7 | 1.5 | 1.5 | 1.5 | 1.4 | 1.7 | 1.54 | 0.09 | – |
| Pb | 23 | 22 | 25 | 23 | 22 | 25 | 23 | 23 | 22 | 25 | 23.25 | 1.09 | – |
| Bi | < 0.4 | 1.9 | 0.9 | 1.3 | 0.8 | 0.7 | 1.5 | 1.3 | 0.7 | 1.9 | 1.2 | 0.4 | – |
| Th | 6.7 | 7 | 6.8 | 6.3 | 6.4 | 6.7 | 7.2 | 6.7 | 6.3 | 7.2 | 6.73 | 0.27 | 17 |
| U | 10.6 | 12.6 | 10.2 | 11.1 | 15.8 | 13.3 | 12.1 | 10.1 | 10.1 | 15.8 | 11.98 | 1.81 | 3 |
| Hf | 2 | 2.2 | 2.2 | 2.1 | 2.3 | 2.3 | 2.1 | 2.1 | 2 | 2.3 | 2.16 | 0.1 | 3 |
| K/Rb | 144.8 | 147.22 | 144.16 | 144.37 | 131.96 | 149.02 | 147.66 | 148.63 | 131.96 | 149.02 | 144.73 | 5.15 | – |
| Rb/Sr | 3.58 | 3.63 | 3.49 | 3.89 | 3.88 | 3.41 | 4.15 | 4.07 | 3.41 | 4.15 | 3.76 | 0.26 | – |
| $R_1$ | 3747 | 3792 | 3673 | 3779 | 3703 | 3786 | 3752 | 3762 | 3673 | 3792 | 3749.25 | 39.04 | – |
| $R_2$ | 227.93 | 232.85 | 225.08 | 226.89 | 234.99 | 249.56 | 245.03 | 242.96 | 225.08 | 249.56 | 235.66 | 8.60 | – |
| A | 30.56 | 29.23 | 19.64 | 35.73 | 25.92 | 21.62 | 28.6 | 25.43 | 19.64 | 35.73 | 27.09 | 4.78 | – |
| B | 28.99 | 27.02 | 25.48 | 28.91 | 27.01 | 30.28 | 33.5 | 33.42 | 25.48 | 33.5 | 29.33 | 2.76 | – |
| A/CNK | 1.38 | 1.37 | 1.28 | 1.43 | 1.34 | 1.31 | 1.37 | 1.35 | 1.28 | 1.43 | 1.35 | 0.04 | – |
| A/NK | 1.52 | 1.53 | 1.41 | 1.57 | 1.49 | 1.49 | 1.52 | 1.51 | 1.41 | 1.57 | 1.51 | 0.04 | – |

$R_1 = 4Si - 11(Na + K) - 2(Fe + Ti)$; $R_2 = 6Ca + Mg + Al$; $A = Al - (K + Na + 2Ca)$; $B = Fe + Mg + Ti$; alumina saturation index: molar $A/CNK = Al_2O_3/(CaO + Na_2O + K_2O)$; molar $A/NK = Al_2O_{3/}(Na_2O + K_2O)$.

According to the classification proposed in [102], the CG reveals a magnesian association in its *Fe*-number (or Fe*) versus $SiO_2$ plot (adapted from [103]) and calc–alkalic to alkali–calcic association based on a modified alkali–lime index (MALI, modified from [104]) (Figure 6a,b respectively). The aluminum saturation index shows a moderated peraluminous character with a molar A/CNK index ranging from 1.28 to 1.43 [105]. The CG plots are included in the granite sensu stricto field in the $R_1$–$R_2$ diagram [106] (Figure 6c). The CG plots in the peraluminous field with muscovite > biotite are presented in the B–A diagram [107] (Figure 6d), displaying affinities with the post-orogenic granites defined in [108]. Capinha granite is geochemically differentiated, as evidenced by its high $SiO_2$ and low CaO, FeO, MgO and $TiO_2$ contents; it is also enriched in Sn and W.

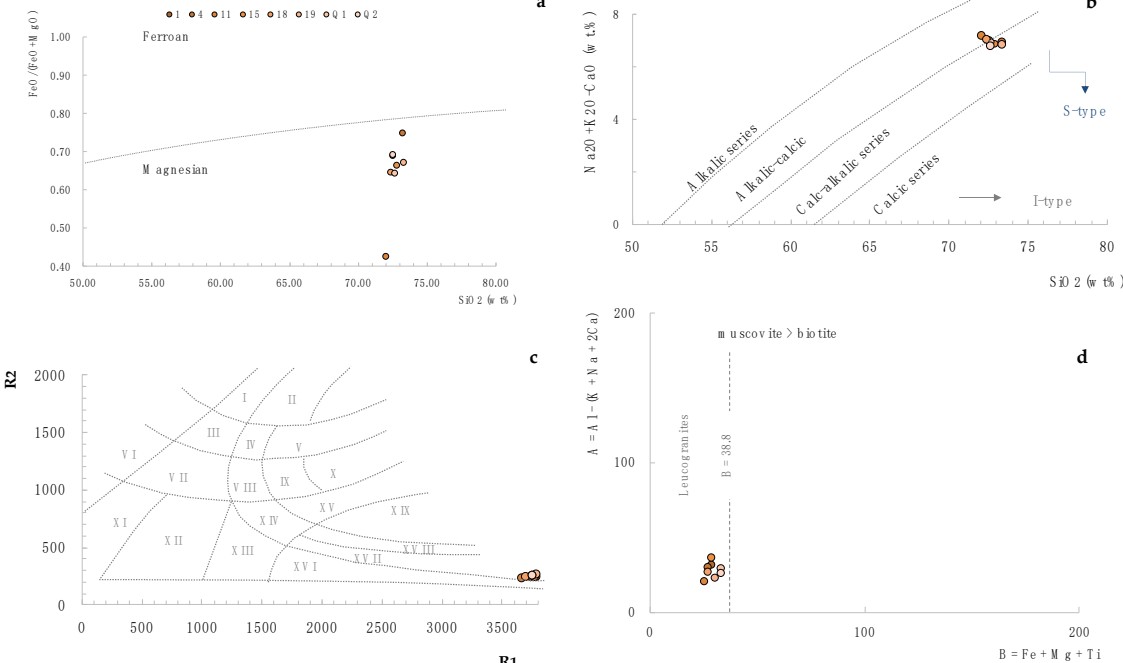

**Figure 6.** Geochemical classification of Capinha granite based on the major elements. (**a**) $FeO^{(Tot)}/FeO^{(Tot)}$ + MgO versus $SiO_2$ [101]; (**b**) modified alumina–lime index (MALI) versus $SiO_2$ classification [102]; (**c**) multicationic $R_1$–$R_2$ classification plot from [106]; (**d**) B–A diagram [107] showing the CG samples plotted in the leucogranites of the muscovite > biotite field.

**Table 5.** Rare earth elements (REE, in ppm) of Capinha granite. The REE were normalized to chondrite according to [109]. $REE^{(Tot)}$—the sum of all rare earth elements; $LREE^{(Tot)}$—the sum of all light rare earth elements; $HREE^{(Tot)}$—the sum of all heavy rare earth elements; $_N$—the normalized ratio; Min—minimum; Max—maximum; $\bar{x}$—average; σ—standard deviation

| REE | Capinha Granite | | | | | | | | Min | Max | $\bar{x}$ | σ |
|---|---|---|---|---|---|---|---|---|---|---|---|---|
| | 1 | 4 | 11 | 15 | 18 | 19 | Q1 | Q2 | | | | |
| La | 15.8 | 15.3 | 14.1 | 13.7 | 14.2 | 14.1 | 14.9 | 14.3 | 13.7 | 15.8 | 14.55 | 0.67 |
| Ce | 32.6 | 32.4 | 29.8 | 29.1 | 30.5 | 30.3 | 32 | 30.2 | 29.1 | 32.6 | 30.86 | 1.21 |
| Pr | 3.71 | 3.78 | 3.47 | 3.26 | 3.54 | 3.58 | 3.64 | 3.51 | 3.26 | 3.78 | 3.56 | 0.15 |
| Nd | 14.1 | 14.3 | 13.8 | 12.9 | 13.3 | 12.7 | 14.4 | 13.5 | 12.7 | 14.4 | 13.63 | 0.59 |
| Sm | 3.2 | 3.4 | 3 | 2.9 | 3 | 2.9 | 3 | 3 | 2.9 | 3.4 | 3.05 | 0.16 |
| Eu | 0.48 | 0.56 | 0.53 | 0.54 | 0.49 | 0.46 | 0.44 | 0.51 | 0.44 | 0.56 | 0.50 | 0.04 |
| Gd | 2.9 | 3.3 | 2.8 | 2.9 | 3.3 | 2.6 | 2.7 | 2.9 | 2.6 | 3.3 | 2.93 | 0.24 |
| Tb | 0.4 | 0.5 | 0.4 | 0.5 | 0.5 | 0.4 | 0.4 | 0.4 | 0.4 | 0.5 | 0.44 | 0.05 |
| Dy | 2.5 | 2.8 | 2.3 | 2.3 | 2.5 | 2.3 | 2.4 | 2.2 | 2.2 | 2.8 | 2.41 | 0.18 |
| Ho | 0.4 | 0.5 | 0.4 | 0.4 | 0.4 | 0.4 | 0.4 | 0.4 | 0.4 | 0.5 | 0.41 | 0.03 |
| Er | 1.1 | 1.2 | 1.1 | 1.1 | 1.2 | 1 | 1 | 1 | 1 | 1.2 | 1.09 | 0.08 |
| Tm | 0.16 | 0.16 | 0.16 | 0.18 | 0.16 | 0.14 | 0.16 | 0.14 | 0.14 | 0.18 | 0.16 | 0.01 |
| Yb | 1 | 1 | 0.9 | 1.1 | 0.9 | 0.9 | 0.9 | 0.9 | 0.9 | 1.1 | 0.95 | 0.07 |
| Lu | 0.13 | 0.14 | 0.13 | 0.14 | 0.14 | 0.11 | 0.14 | 0.12 | 0.11 | 0.14 | 0.13 | 0.01 |

| REE | Capinha Granite | | | | | | | | Min | Max | $\bar{x}$ | σ |
|---|---|---|---|---|---|---|---|---|---|---|---|---|
| | 1 | 4 | 11 | 15 | 18 | 19 | Q1 | Q2 | | | | |
| REE(Tot) | 78.48 | 79.34 | 72.89 | 71.02 | 74.13 | 71.89 | 76.48 | 73.08 | 71.02 | 79.34 | 74.66 | 2.89 |
| LREE(Tot) | 69.41 | 69.18 | 64.17 | 61.86 | 64.54 | 63.58 | 67.94 | 64.51 | 61.86 | 69.41 | 65.65 | 2.63 |
| HREE(Tot) | 8.59 | 9.6 | 8.19 | 8.62 | 9.1 | 7.85 | 8.1 | 8.06 | 7.85 | 9.6 | 8.51 | 0.55 |
| (La/Lu)$_N$ | 12.62 | 11.34 | 11.26 | 10.16 | 10.53 | 13.31 | 11.05 | 12.37 | 10.16 | 13.31 | 11.58 | 1.02 |
| (La/Sm)$_N$ | 3.11 | 2.83 | 2.96 | 2.97 | 2.98 | 3.06 | 3.13 | 3 | 2.83 | 3.13 | 3.01 | 0.09 |
| (Gd/Lu)$_N$ | 2.77 | 2.93 | 2.68 | 2.57 | 2.93 | 2.94 | 2.4 | 3 | 2.4 | 3 | 2.78 | 0.20 |
| (Eu/Eu)$_N$ | 0.24 | 0.25 | 0.28 | 0.28 | 0.24 | 0.25 | 0.23 | 0.26 | 0.23 | 0.28 | 0.25 | 0.02 |

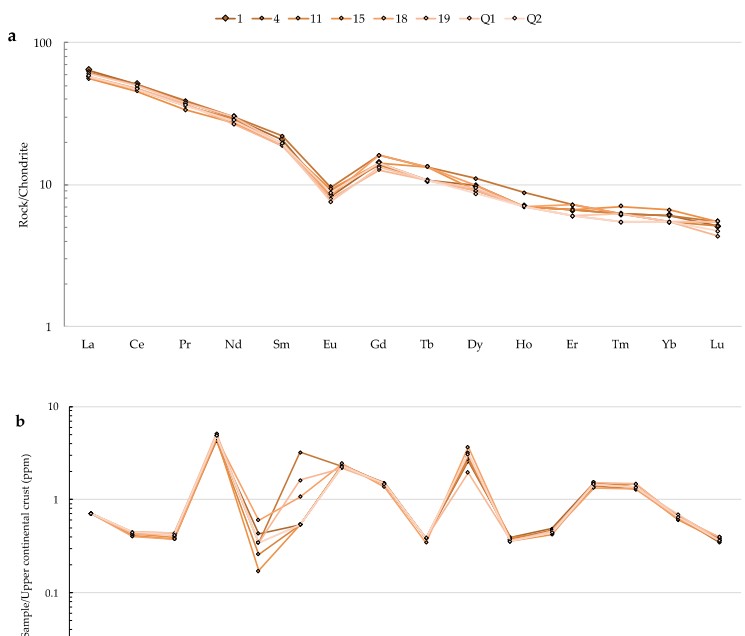

**Figure 7.** (**a**) Chondrite-normalized REE distribution patterns from the CG. Normalization values are from [109]; (**b**) multi-element diagram normalized to the upper continental crust (UCC) for CG samples. Normalization values are from [110].

The chondrite normalized REE patterns [109] for the CG showed very similar spectra profiles (Table 5), demonstrating a strong overlap in light rare earth elements (LREE) compared to heavy rare earth elements (HREE) (Figure 7). The LREE and HREE are slightly fractionated here, corresponding to (La/Sm)$_N$ = 2.83 to 3.13 and (Gd/Lu)$_N$ = 2.4 to 3, respectively. The CG is enriched in LREE(Tot) (61.86–69.41 ppm) compared to HREE(Tot) (7.85–9.6 ppm), showing moderated REE fractionation, with (La/Lu)$_N$ ranging from 10.16 to 13.31 and a small negative anomaly in Eu (Eu/Eu*)$_N$, ranging between 0.23 and 0.28.

Multi-element diagram normalized to the upper continental crust [110] (Figure 7b) show similar patterns and a remarkable overlap of all CG samples, with positive Cr, Rb, Zr, Cs, Ta and Pb anomalies and negative Sc, Ti, V, Co, Nb, Ba, La, Th and Hf anomalies. Ni displays a wide distribution.

5.1.4. Comparative Study between Capinha, Peroviseu–Seia, Belmonte–Covilhã and Fráguas Granites

A geochemical comparative study between CG and PS and BC and Fráguas granites from the FSEC region was carried out to delineate the possible evolutionary trend of the granites and to investigate their metallogenic potential.

In the Fundão–Serra da Estrela–Capinha region, the Alpine orogeny triggered crustal uplift, thereby developing the "Serra da Estrela Horst" (SEH, currently ~2000 m high). The construction of the SEH and subsequent erosion exposed deep structural levels to the surface.

The Fundão–Serra da Estrela–Capinha region consists of three distinct erosional levels according to their lithological and mineralization characteristics:

(1) The erosion level of the Serra da Estrela (~2000 m above sea level), characterized by the occurrence of essentially granitic rocks; the metasedimentary rocks were completely eroded, so there is no evidence of ore deposits.

(2) The erosion level of Capinha (527 m above sea level), characterized by the occurrence of different granitic facies and low metamorphic metasedimentary rocks, whose contact is marked by hornfels; the occurrence of W and W (Sn) mineralizations is evident in the exo-contact between granites/metasediments and intragranitic;

(3) The erosion level of Fundão (depression at SW of SEH, 500 m above sea level) between Panasqueira and Argemela, characterized by the occurrence of essentially metasedimentary rocks with no outcropping of granitic rocks; W (Sn) occurrences are well preserved and hosted in the metasedimentary rocks, which are spatially associated with non-outcropping granitic rocks, e.g., Panasqueira. Exploration of the Panasqueira deposit in underground works showed the presence of a greisen cupola (1st evidence found in 1946), which is related to non-outcropping parental granite. A drilling campaign performed in 1953 cut greisenized granite at a ~190 m depth (ca. 450 m above sea level), which is located at about 1550 m lower than the highest point of Serra da Estrela and at a similar height to the Capinha massif (527 m above sea level).

Major and Trace Elements

The major and trace element contents of the granites from the FSEC region are presented in Table 6. All the rocks have a granitic composition, with $SiO_2$ values varying from 67.12 wt% to 74.13 wt%; $TiO_2$ values of ~0.08 wt%–0.72 wt%; $Fe_2O_3^{(Tot)}$ values of ~1.14 wt%–4.08 wt%, showing moderately high alkali contents; $Al_2O_3$ values of ~14.04 wt%–15.76%; $Na_2O$ values of ~2.49 wt%–3.47 wt%; and $K_2O$ values of ~4.26 wt%–5.27 wt%, with low quantities of CaO (~0.39 wt%–1.47 wt%) and MgO (~0.17 wt%–1.3 wt%). Fráguas granite is the richest in $SiO_2$ (73.36 wt%–74.13 wt%), BC is the richest in $K_2O$ (5.04 wt%–5.12 wt%) and PS is the richest in $Fe_2O_3^{(Tot)}$ (2.47 wt%–4.08 wt%). $TiO_2$ decreases successively from PS–BC–Capinha–Fráguas. The $P_2O_5$ content is higher in Fráguas and Capinha granites.

The CG and BC are magnesian, and the PS and Fráguas are ferroan [102]. All the granites display associations with the high-K calc–alkaline affinity in the $K_2O$ versus $SiO_2$ binary plot (Figure 8a). CG, BC and PS are peraluminous, with molecular A/CNK ratios ranging from 1.28 to 1.43, 1.15 to 1.32 and 1.17 to 1.36, respectively. The Fráguas granite plots in the metaluminous field with A/CNK range from 0.92 to 1 (Figure 8b).

**Table 6.** Major (wt%) and trace element (ppm) data of Capinha were obtained in this study. The whole rock geochemical data of PS, BC and Fráguas were compiled from [28,70,83]. n.d.—not determined; LOI—lost on ignition; n—number of samples; Min—minimum; Max—maximum; $\bar{x}$—average; σ—standard deviation.

| Element | Peroviseu–Seia (*n* = 5) | | | | Belmonte–Covilhã (*n* = 3) | | | | Fráguas (*n* = 3) | | | | Capinha (*n* = 8) | | | |
|---|---|---|---|---|---|---|---|---|---|---|---|---|---|---|---|---|
| | Min | Max | $\bar{x}$ | σ | Min | Max | $\bar{x}$ | σ | Min | Max | $\bar{x}$ | σ | Min | Max | $\bar{x}$ | σ |
| SiO$_2$ | 67.12 | 72.4 | 70.06 | 1.81 | 71.14 | 73.31 | 72.13 | 0.9 | 73.36 | 74.13 | 73.68 | 0.33 | 72.07 | 73.41 | 72.76 | 0.42 |
| Al$_2$O$_3$ | 14.04 | 15.76 | 14.9 | 0.63 | 14.29 | 14.86 | 14.65 | 0.25 | 14.34 | 14.89 | 14.64 | 0.23 | 14.33 | 15.1 | 14.8 | 0.24 |
| Fe$_2$O$_3$ | 0.4 | 0.79 | 0.62 | 0.13 | 0.42 | 0.68 | 0.55 | 0.11 | 0.19 | 0.28 | 0.22 | 0.04 | 0.55 | 1.69 | 1.01 | 0.33 |
| MnO | 0.04 | 0.05 | 0.05 | 0.005 | 0.03 | 0.04 | 0.04 | 0.005 | 0.05 | 0.07 | 0.06 | 0.01 | 0.03 | 0.05 | 0.04 | 0.01 |
| FeO | 1.86 | 2.96 | 2.36 | 0.46 | 1.09 | 1.65 | 1.42 | 0.24 | 0.85 | 1.52 | 1.17 | 0.27 | 0.30 | 1.50 | 1.04 | 0.34 |
| MgO | 0.48 | 1.3 | 0.77 | 0.3 | 0.3 | 0.65 | 0.45 | 0.15 | 0.17 | 0.22 | 0.19 | 0.02 | 0.41 | 0.67 | 0.52 | 0.09 |
| CaO | 0.87 | 1.47 | 1.12 | 0.2 | 0.67 | 0.9 | 0.75 | 0.1 | 0.39 | 0.48 | 0.43 | 0.04 | 0.50 | 0.73 | 0.6 | 0.06 |
| Na$_2$O | 2.49 | 2.89 | 2.67 | 0.16 | 2.74 | 3.47 | 3.09 | 0.3 | 2.97 | 3.24 | 3.13 | 0.11 | 2.93 | 3.32 | 3.09 | 0.13 |
| K$_2$O | 4.84 | 5.27 | 5.05 | 0.16 | 5.04 | 5.12 | 5.09 | 0.03 | 4.42 | 4.73 | 4.6 | 0.13 | 4.26 | 4.5 | 4.4 | 0.09 |
| TiO$_2$ | 0.33 | 0.72 | 0.48 | 0.14 | 0.16 | 0.32 | 0.26 | 0.07 | 0.08 | 0.13 | 0.11 | 0.02 | 0.21 | 0.23 | 0.22 | 0.01 |
| P$_2$O$_5$ | 0.2 | 0.36 | 0.29 | 0.06 | 0.22 | 0.29 | 0.26 | 0.03 | 0.28 | 0.4 | 0.34 | 0.05 | 0.33 | 0.38 | 0.36 | 0.01 |
| F | 0.1 | 0.25 | 0.16 | 0.06 | 0.12 | 0.17 | 0.15 | 0.02 | 0.12 | 0.49 | 0.31 | 0.15 | n.d. | n.d. | n.d. | n.d. |
| LOI | 0.98 | 2.18 | 1.36 | 0.45 | 0.93 | 1.46 | 1.11 | 0.25 | 1.06 | 1.21 | 1.12 | 0.07 | 0.98 | 1.56 | 1.22 | 0.18 |
| Total | 99.73 | 99.99 | 99.87 | 0.1 | 99.73 | 100.05 | 99.94 | 0.15 | 99.87 | 100.14 | 100 | 0.11 | 98.75 | 100.74 | 100.06 | 0.61 |
| Nb | 11 | 18 | 14.4 | 3.01 | 14 | 21 | 17 | 2.94 | 17 | 23 | 20.67 | 2.62 | 9 | 10 | 9.88 | 0.33 |
| Zn | 48 | 66 | 59 | 7.25 | 62 | 71 | 66.5 | 4.5 | 35 | 71 | 53 | 18 | 50 | 90 | 62.5 | 11.99 |
| Sn | 11 | 18 | 13.4 | 2.42 | 21 | 38 | 28 | 7.26 | 39 | 76 | 62 | 16.39 | 9 | 24 | 16.88 | 5.09 |
| Li | 108 | 600 | 241.6 | 181.84 | 193 | 400 | 289.33 | 85.11 | 688 | 1100 | 829 | 191.68 | n.d. | n.d. | n.d. | n.d. |
| Zr | 112 | 208 | 164.4 | 34.06 | 76 | 145 | 112.67 | 28.34 | 29 | 65 | 48.33 | 14.82 | 71 | 79 | 76.5 | 2.24 |
| Y | 19 | 30 | 23.6 | 3.77 | 15 | 22 | 18 | 2.94 | 10 | 19 | 14.33 | 3.68 | 11 | 13 | 12.25 | 0.83 |
| Sr | 66 | 152 | 96 | 35.2 | 51 | 60 | 55 | 3.74 | 21 | 33 | 27 | 4.9 | 61 | 74 | 67.38 | 4.58 |
| Ba | 226 | 600 | 368.4 | 146.13 | 172 | 263 | 208.33 | 39.35 | 46 | 82 | 61.33 | 15.17 | 239 | 263 | 248.5 | 8.72 |
| Rb | 228 | 299 | 274 | 26.1 | 190 | 351 | 290 | 71.28 | 465 | 607 | 523.33 | 60.68 | 239 | 268 | 252.5 | 8.26 |
| Th | 12 | 27 | 18.25 | 6.14 | 12 | 13 | 12.5 | 0.5 | 7 | 7 | 7 | 0 | 6.3 | 7.2 | 6.73 | 0.27 |
| W | n.d. | n.d. | 3 | n.d. | n.d. | n.d. | 3 | n.d. | n.d. | n.d. | 2 | n.d. | 6 | 12 | 9.63 | 1.93 |
| K/Rb | 136.88 | 182.42 | 154.38 | 15.19 | 119.21 | 222.84 | 157.08 | 46.68 | 63.73 | 84.45 | 73.96 | 8.46 | 131.96 | 149.02 | 144.73 | 5.15 |
| Fe$_2$O$_3$$^{(Tot)}$ | 2.47 | 4.08 | 3.24 | 0.62 | 1.63 | 2.38 | 2.13 | 0.35 | 1.14 | 1.88 | 1.53 | 0.31 | 2.02 | 2.25 | 2.17 | 0.07 |
| Rb/Sr | 1.5 | 4.53 | 3.27 | 1.16 | 3.52 | 6.88 | 5.29 | 1.38 | 18.39 | 22.14 | 19.66 | 1.76 | 3.41 | 4.15 | 3.76 | 0.25 |
| A/CNK | 1.17 | 1.36 | 1.26 | 0.07 | 1.15 | 1.32 | 1.23 | 0.07 | 0.92 | 1 | 0.96 | 0.03 | 1.28 | 1.43 | 1.35 | 0.04 |
| A/NK | 1.39 | 1.66 | 1.51 | 0.1 | 1.31 | 1.48 | 1.39 | 0.07 | 1.39 | 1.5 | 1.45 | 0.05 | 1.41 | 1.57 | 1.5 | 0.04 |

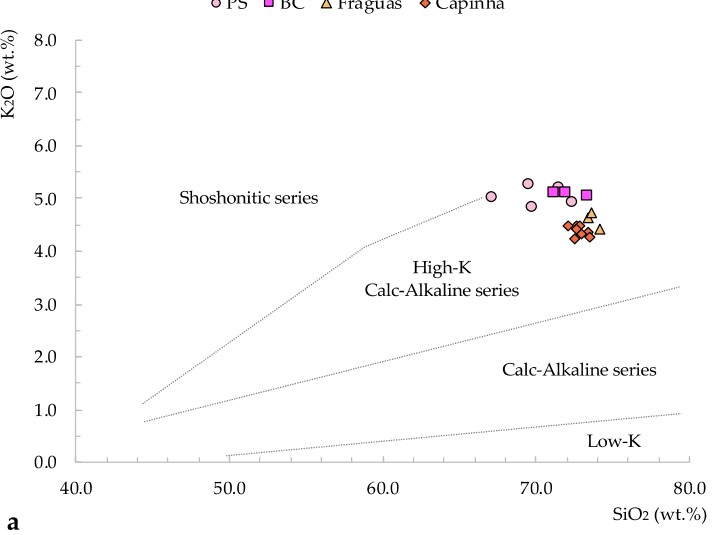

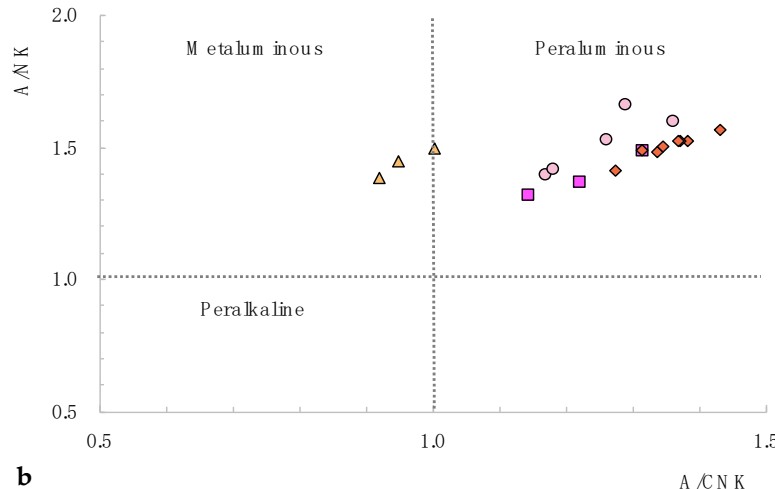

**Figure 8.** (**a**) K$_2$O (wt%) versus SiO$_2$ (wt%) plot showing the strong association between the granites from Fundão–Serra da Estrela–Capinha (FSEC) and the high-K calc–alkaline series; (**b**) A/NK versus A/CNK plot [105] displaying the aluminum saturation of the FSEC granites.

The K/Rb ratio is a good indicator of magmatic differentiation [111], as Rb tends to concentrate in evolved residual melts due to its high incompatibility degree with the early formed mineral phases. Instead, K is included in the crystalline structures of alkali minerals and consequently decreases more quickly during magmatic differentiation. In response, the K/Rb ratio decreases with an increase in magmatic differentiation ranging from 63.73 to 222.84. The variation diagrams for SiO$_2$, TiO$_2$, Al$_2$O$_3$, Fe$_2$O$_3$$^{(Tot)}$, CaO, Na$_2$O, K$_2$O, P$_2$O$_5$, Ba and Sn versus K/Rb reveal that each granite experiences its own differentiation trend. However, the CG the samples are mostly clustered (Figure 9). The variation diagrams show that TiO$_2$, Fe$_2$O$_3$$^{(Tot)}$, CaO, K$_2$O and Ba decrease with a decrease in the K/Rb ratio. On the other hand, the content of SiO$_2$, Al$_2$O$_3$, Na$_2$O, P$_2$O$_5$ and Sn increases through magmatic differentiation. These variation diagrams also demonstrate that Fráguas granite is the most evolved facies, displaying K/Rb ratios lower than 100.

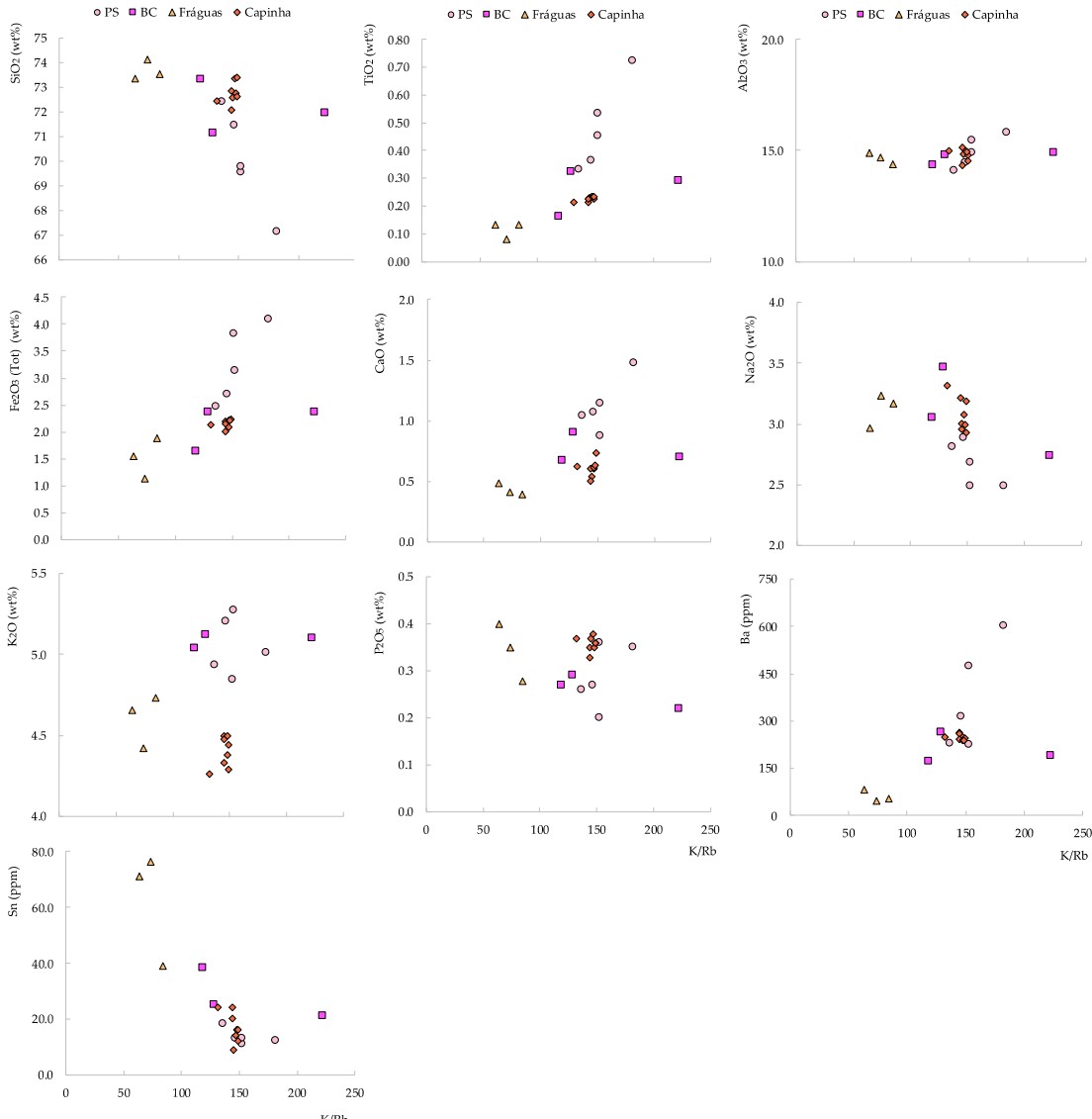

**Figure 9.** Variation diagrams showing major ($SiO_2$, $TiO_2$, $Al_2O_3$, $Fe_2O_3^{(Tot)}$, CaO, $Na_2O$, $K_2O$ and $P_2O_5$) and trace element (Ba and Sn) distributions in the granites of the CG, Fráguas, BC and PS. The differentiation index (K/Rb) is used in all graphs. A decrease in K/Rb represents an increase in magmatic fractionation.

The Rb/Sr ratio versus Sn, as an indicator of the degree of magmatic differentiation, showed that the Sn increases with an increase in Rb/Sr for the Bc and PS granites. On the other hand, the Fráguas and Capinha granites display constant Rb/Sr values with an increase in Sn (Figure 10a). The Sr contents remained essentially unchanged and low (<20).

The magmatic differentiation degree is also shown in the Rb–Sr–Ba ternary diagram [111], where most the granites are plotted in the strongly differentiated granite field. The high degree of differentiation of the Fráguas granite is supported by its high $SiO_2$ contents; enrichment in incompatible elements such as Li, Rb, Sn and Nb; and low quantities of Ba, Sr, Zr and Y. The fractionation process continuously begins with the least differentiated facies, the PS and ends with the most evolved facies, Fráguas granite (Figure 10b). This fractionation is gradual and characterized by a quick decrease in Ba, which replaces K in the structure of the K feldspar, while the Rb contents relatively increase their concentrations in the residual melts [28,112].

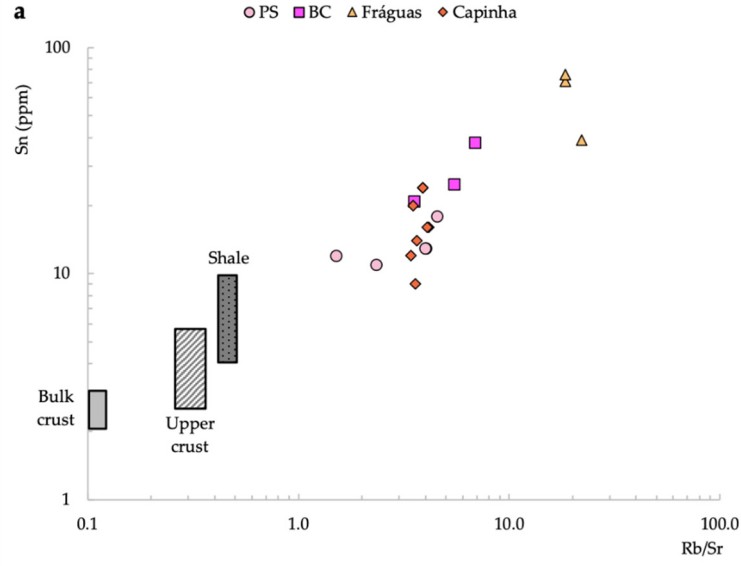

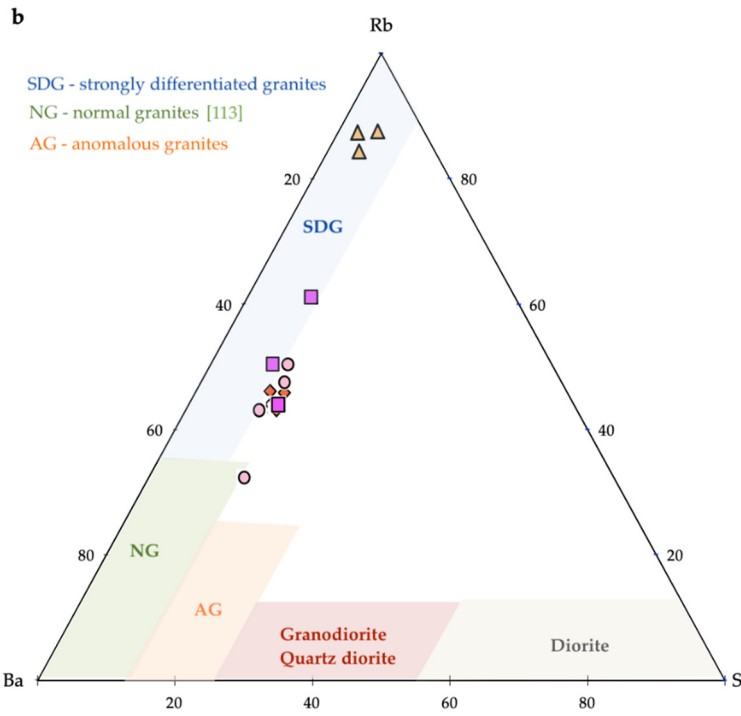

**Figure 10.** (**a**) Correlation of Rb/Sr versus Sn for the granites in the study (logarithmic scale). The fields representative of bulk crust, upper crust and shale composition were taken from [28]; (**b**) Rb–Ba–Sr ternary diagram showing the evolution trend from the differentiation sequence between the diorite and strongly evolved granites [113] with the representation of the granites in the study clustered in the strongly differentiated granite field (blue field).

Rare Earth Elements

Table 7 presents rare earth element data of the studied granites. Chondrite-normalized REE patterns of the granites display very similar profiles, although the negative anomaly in (Eu/Eu)$_N$ is more pronounced in the Fráguas granite (Figure 11). The BC and PS profiles strongly overlap in the LREE (110.79 ppm and 113.3 ppm, respectively) compared to the HREE, where the PS is slightly enriched (14.2 ppm). The Capinha granite is more depleted in REE$^{(Tot)}$ (74.67 ppm) than BC (REE$^{(Tot)}$ ~123.38 ppm) and PS (REE$^{(Tot)}$ ~128.07 ppm), showing the most evolved REE pattern. The Capinha

and Fráguas profiles are parallel in the LREE; however, Capinha is more impoverished in HREE than Fráguas, which is marked by a high REE fractionation with $(La/Yb)_N$ values of 10.34 and 7.19, respectively. The negative anomaly in $Eu_N$, related to the fractionation of feldspar, is more evident in the Fráguas granite $((Eu/Eu^*)_N \sim 0.09)$ compared to the other granites. Therefore, the CG exhibits the fewest marked negative anomalies in $Eu_N$ (typical of post-orogenic granites).

**Table 7.** Rare earth element (ppm) data for CG were obtained in this study. The whole rock geochemical data of PS, BC and Fráguas were compiled from [28,83,114]. The REE (rare earth elements) were normalized to chondrite according to [108]. n, number of samples; $REE^{(Tot)}$, sum of all rare earth elements; $LREE^{(Tot)}$, sum of all light rare earth elements; $HREE^{(Tot)}$, some of all heavy rare earth elements; $_N$, normalized ratio.

| Rare Earth Element | Peroviseu–Seia (PS) *n* = 5 | Belmonte–Covilhã (BC) *n* = 3 | Fráguas *n* = 3 | Capinha *n* = 8 |
|---|---|---|---|---|
| La | 23.7 | 25.8 | 10.65 | 14.55 |
| Ce | 53.8 | 50.7 | 24.4 | 30.86 |
| Pr | 6.4 | 5.99 | 2.9 | 3.56 |
| Nd | 24.2 | 23.7 | 10.8 | 13.63 |
| Sm | 5.2 | 4.6 | 2.6 | 3.05 |
| Eu | 0.57 | 0.65 | 0.15 | 0.5 |
| Gd | 4.3 | 4.04 | 2.45 | 2.93 |
| Tb | 0.73 | 0.6 | 0.42 | 0.44 |
| Dy | 4.1 | 3.13 | 2.65 | 2.41 |
| Ho | 0.77 | 0.68 | 0.45 | 0.41 |
| Er | 1.9 | 1.54 | 1.15 | 1.09 |
| Tm | 0.3 | 0.23 | 0.17 | 0.16 |
| Yb | 11.8 | 1.5 | 1 | 0.955 |
| Lu | 0.3 | 0.22 | 0.15 | 0.13 |
| $LREE^{(Tot)}$ | 113.3 | 110.79 | 51.35 | 65.65 |
| $HREE^{(Tot)}$ | 14.2 | 11.94 | 8.44 | 8.52 |
| $REE^{(Tot)}$ | 128.07 | 123.38 | 59.94 | 74.67 |
| $(La/Sm)_N$ | 2.87 | 3.53 | 2.58 | 3.00 |
| $(Gd/Yb)_N$ | 1.93 | 2.18 | 1.98 | 2.49 |
| $(La/Yb)_N$ | 8.89 | 11.61 | 7.19 | 10.34 |
| $(Eu/Eu^*)_N$ or $Eu_N$ | 0.18 | 0.23 | 0.09 | 0.25 |

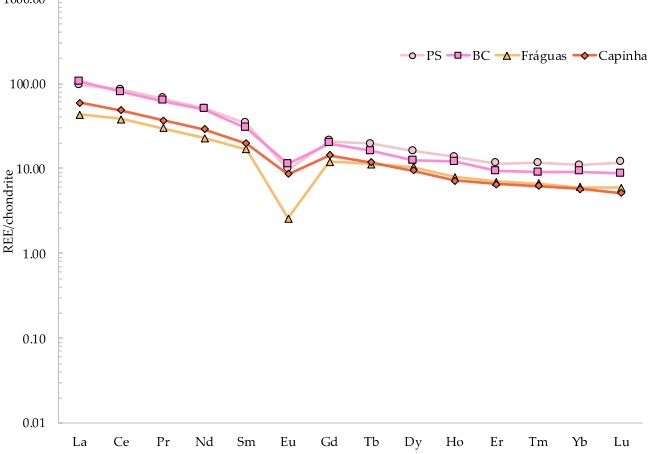

**Figure 11.** Rare earth element profiles for the PS, BC, CG and Fráguas granites (chondrite-normalized according to [109].

The REE patterns of PS and BC are subparallel. PS, moreover, is richer in all REE and has a lower $(La/Yb)_N$ ratio (8.89) than BC (11.61). The REE patterns of BC are similar to those of PS in Sm and Gd, exhibiting a higher $Eu_N$ (0.23) value than PS (0.18) and suggesting that BC and PS are not related to each other. The CG, PS and BC are subparallel, where CG is poorer in REE (74.67 ppm) than BC (123.38 ppm) and PS (128.07 ppm). However, the CG flattened profile characterized by high $Eu_N$ (0.25) indicates a distinct geological setting for its origin. The REE patterns of Fráguas and CG are subparallel in the LREE, whereas the REE spectra of Fráguas intercept those of CG in HREE, suggesting that Fráguas and CG are not related to each other. Considering these observations, the REE spectra suggest that PS, BC, CG and Fráguas are independent.

## 5.2. Anisotropy of Magnetic Susceptibility

The values of $K_m$, $P_{para}$, $T$, the orientation of $K_1$ and $K_3$ and the 95% confidence angles (E12, E23, E31) of the $K_1$, $K_2$, $K_3$ axes of each sampling site are summarized in Table 8.

### 5.2.1. Scalar Parameters

Bulk Magnetic Susceptibility ($K_m$)

The $K_m$ magnitudes at each sampling site display a constrained range, covering one order of magnitude, which is typical of a paramagnetic contribution, ranging from 17.4 to 89.47 μS.I. (=73.4 μS.I.) (Table 8). The spatial distribution of $K_m$ in the whole granite (Figure 12a) showed a homogeneous zonation analogous to the statistical data, which show that 76.67% of the total measurements have a $K_m$ higher than 70 μS.I. (see the frequency histogram Figure 12a). The lowest values are recorded in the eastern and western granite peripheries (sampling sites 17 and 8, respectively), which coincide with both the approximate axis of susceptibility symmetry and the granite long axis (NNE–SSW oriented). The highest values are concentrated in the northern zone. Although the granite is strongly homogeneous in its mineralogy and susceptibility, only a slight correlation between the granite mineralogy and susceptibility was observed. In zones where the amount of biotite is higher than muscovite, magnetic susceptibility increases. Since the biotite and muscovite are magmatic, magnetic susceptibility is strictly magmatic in origin.

**Table 8.** AMS data for the 30 sampling sites in Capinha granite (CG). *n*, number of specimens, $K_m$, bulk magnetic susceptibility expressed in µS.I. (or $10^{-6}$ SI) units; $P_{para}$, magnitude of the paramagnetic anisotropy in%; *T*, shape parameter; σ, standard deviation; SE, standard error; $K_1$, $K_3$, orientation of average principal maximum, medium and minimum susceptibility axes (D, dip direction; I, dip); E12, E23, E31, eigenvalues [115], Min, minimum; Max, maximum; $\bar{x}$, average; n.d., not determined.

| Site | *n* | $K_m$ | σ | $P_{para}$ | *T* | σ | Magnetic Lineation | | $K_3D$ | $K_3I$ | Magnetic Foliation | E12 | E23 | E31 | SE |
|---|---|---|---|---|---|---|---|---|---|---|---|---|---|---|---|
| | | | | | | | $K_1D$ | $K_1I$ | | | | | | | |
| 1 | 7 | 89.6 | 9.4 | 4.44 | −0.17 | 0.3 | 194 | 43 | 297 | 13 | 27° NE; 77° E | 1.52 | 2.86 | 0.99 | 0.04 |
| 2 | 7 | 79.9 | 4.71 | 1.39 | 0.11 | 0.52 | 139 | 35 | 272 | 45 | N2° E; 45° E | 10.3 | 8.61 | 3.07 | 0.05 |
| 3 | 12 | 72.1 | 5.42 | 1.6 | −0.04 | 0.15 | 302 | 10 | 190 | 65 | N280° E; 25° NE | 7.24 | 7.36 | 3.88 | 0.06 |
| 4 | 8 | 71 | 4.58 | 1.77 | 0.69 | 0.15 | 42 | 2 | 265 | 87 | N355° E; 3° E | 18.94 | 3.68 | 3.1 | 0.06 |
| 5 | 7 | 79.5 | 3.96 | 0.85 | −0.37 | 0.44 | 263 | 7 | 62 | 83 | N152° E; 7° SW | 4.29 | 14.6 | 2.86 | 0.04 |
| 6 | 8 | 80.9 | 5.82 | 1.89 | 0.07 | 0.21 | 201 | 15 | 68 | 69 | N158° E; 21° SW | 6.7 | 5.79 | 3.26 | 0.06 |
| 7 | 12 | 77.6 | 4.65 | 1.19 | 0.41 | 0.27 | 255 | 34 | 79 | 56 | N169° E; 34° SW | 12.19 | 4.94 | 2.99 | 0.07 |
| 8 | 7 | 17.4 | 0.88 | 0.61 | 0.47 | 0.21 | 53 | 11 | 259 | 78 | N349° E; 12° E | 39.97 | 17.7 | 13.2 | 0.15 |
| 9 | 9 | 76.6 | 3.93 | 2.31 | 0.55 | 0.3 | 190 | 33 | 336 | 52 | N66° E; 38° SE | 8.04 | 2.22 | 1.59 | 0.04 |
| 10 | 9 | 74.7 | 3.64 | 1.87 | 0.44 | 0.12 | 6 | 6 | 250 | 76 | N340° E; 14° NE | 5.11 | 2.18 | 1.54 | 0.03 |
| 11 | 8 | 69.2 | 2.98 | 1.43 | 0.47 | 0.46 | 262 | 19 | 129 | 63 | N219° E; 27° NW | 21.98 | 8.84 | 4.61 | 0.08 |
| 12 | 7 | 74.5 | 6.46 | 1.19 | 0.06 | 0.52 | 31 | 5 | 138 | 73 | N228° E; 17° NW | 7.74 | 6.44 | 2.54 | 0.04 |
| 13 | 6 | 75.1 | 8.15 | 1.27 | 0.36 | 0.26 | 209 | 4 | 15 | 86 | N105° E; 4° SW | 9.78 | 4.92 | 3.25 | 0.04 |
| 14 | 6 | 78.2 | 4.38 | 2.15 | 0.56 | 0.2 | 321 | 5 | 228 | 25 | N318° E; 65° NE | 7.22 | 1.97 | 1.48 | 0.04 |
| 15 | 9 | 76.9 | 2.78 | 1.71 | 0.33 | 0.39 | 189 | 3 | 66 | 85 | N156° E; 5° SW | 14.98 | 7.17 | 4.81 | 0.08 |
| 16 | 7 | 66.1 | 3.32 | 1.25 | 0.45 | 0.18 | 97 | 7 | 286 | 83 | N16° E; 7° SE | 8.53 | 3.8 | 2.61 | 0.03 |
| 17 | 6 | 61 | 3.2 | 0.65 | 0.04 | 0.66 | 68 | 14 | 330 | 28 | N60° E; 62° SE | 9.58 | 11.1 | 2.83 | 0.05 |
| 18 | 10 | 74.8 | 4.75 | 1.7 | 0.08 | 0.26 | 70 | 26 | 217 | 59 | N307° E; 31° NE | 6.53 | 5.7 | 2.91 | 0.05 |
| 19 | 6 | 87.9 | 3.86 | 2.18 | −0.33 | 0.24 | 156 | 38 | 281 | 36 | N11° E; 54° SE | 3.13 | 7.22 | 2.08 | 0.05 |
| 20 | 9 | 76.2 | 4.2 | 1.36 | 0.5 | 0.23 | 167 | 13 | 33 | 71 | N123° E; 19° SW | 11.47 | 3.04 | 2.2 | 0.03 |
| 21 | 8 | 66.4 | 4.47 | 2.79 | 0.53 | 0.26 | 64 | 8 | 256 | 82 | N166° E; 8° NE | 9.81 | 2.6 | 2.05 | 0.05 |
| 22 | 9 | 78.7 | 7.59 | 2.53 | 0.67 | 0.16 | 185 | 1 | 87 | 82 | N177° E; 8° W | 8.44 | 1.42 | 1.18 | 0.03 |
| 23 | 8 | 72.5 | 3.04 | 1.65 | 0.29 | 0.2 | 246 | 8 | 353 | 64 | N83° E; 26° SE | 6.31 | 3.1 | 2.03 | 0.03 |
| 24 | 7 | 73.4 | 2.06 | 1.89 | 0.53 | 0.2 | 314 | 1 | 45 | 36 | N135° E; 54° SW | 12.64 | 3.67 | 2.87 | 0.05 |
| 25 | 5 | 64.5 | 3.67 | 2.32 | 0.47 | 0.15 | 243 | 4 | 139 | 77 | N49° E; 13° NW | 6.56 | 3.12 | 1.74 | 0.04 |
| 26 | 6 | 79.5 | 9.76 | 2.1 | 0.18 | 0.17 | 196 | 6 | 97 | 55 | N7° E; 35° NW | 3.45 | 2.6 | 1.47 | 0.03 |
| 27 | 6 | 89.7 | 30.7 | 2.22 | 0.46 | 0.27 | 171 | 24 | 316 | 62 | N226° E; 28° SE | 5.2 | 1.5 | 1.02 | 0.02 |
| 28 | 6 | 75 | 6.74 | 2.64 | −0.23 | 0.26 | 12 | 20 | 140 | 60 | N50° E; 30° NW | 4.1 | 11.6 | 2.82 | 0.07 |
| 29 | 6 | 69.7 | 5.39 | 1.78 | 0.27 | 0.6 | 193 | 13 | 296 | 45 | N206° E; 45° SE | 22.13 | 10.4 | 4.67 | 0.08 |
| 30 | 6 | 73.3 | 4.95 | 8.04 | 0.42 | 0.09 | 199 | 5 | 21 | 85 | N111° E; 5° SW | 6.1 | 2.75 | 1.93 | 0.09 |
| *Min* | | 17.4 | 0.88 | 0.61 | −0.37 | 0.09 | | | | | | 1.52 | 1.42 | 0.99 | 0.02 |
| *Max* | | 89.67 | 30.7 | 8.04 | 0.69 | 0.66 | | | | n.d. | | 39.97 | 17.7 | 13.2 | 0.15 |
| σ | | 12.29 | 5.06 | 1.33 | 0.29 | 0.14 | | | | | | 7.48 | 3.99 | 2.16 | 0.03 |
| $\bar{x}$ | | 73.4 | 5.65 | 2 | 0.3 | 0.28 | 166 | 5 | 328 | 85 | N58° E; 5° SE | 10 | 5.8 | 2.9 | 0.05 |

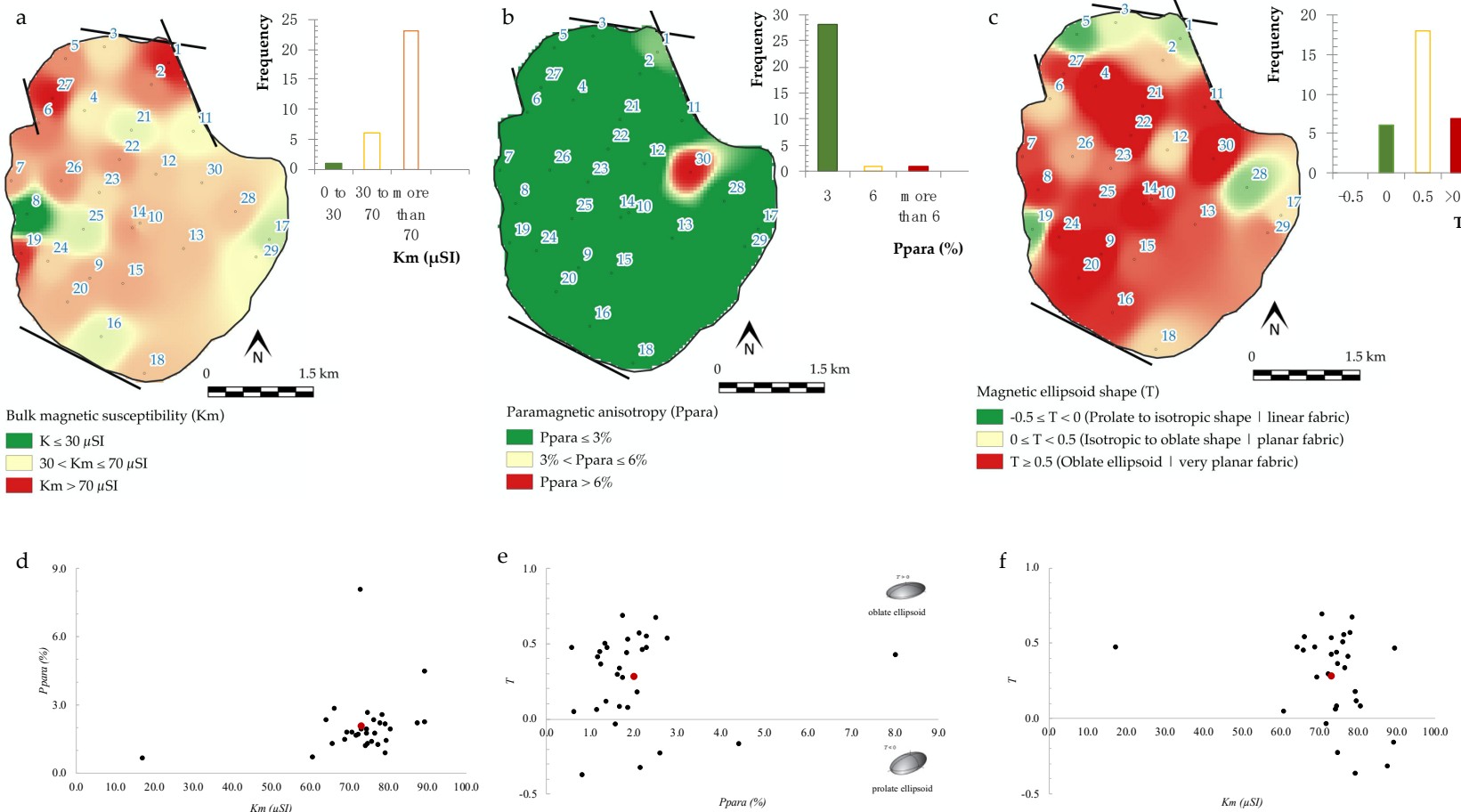

**Figure 12.** Domain and frequency distribution of (**a**) bulk magnetic susceptibility ($K_m$, µS.I.), (**b**) paramagnetic anisotropy percentage ($P_{para}$, %) and (**c**) the magnetic ellipsoid shape parameter ($T$) in Capinha granite. The relationship between the three scalar parameters obtained in AMS studies, namely, (**d**) $K_m$ versus $P_{para}$, (**e**) $P_{para}$ versus $T$ and (**f**) $K_m$ versus $T$ are also presented. Note: the red circle represents the average.

Paramagnetic Anisotropy ($P_{para}$)

The percentage of $P_{para}$ within the granite ranges from 0.61% to 8.04% (Figure 12b). However, the frequency distribution shows that only 6.66% of the total measurements display a $P_{para}$ higher than 3% (see the frequency histogram Figure 12b), demonstrating that $P_{para}$ is relatively homogeneous at the intrusion scale. The highest values are located in the peripheries (at sampling sites 1 and 30), reflecting the low-temperature solid state deformation experienced by the granite during magma solidification. The average value is 2.0% for the magmatic to submagmatic deformation. The charts in Figure 12d,e show the directly proportional relationship between $K_m$ versus $P_{para}$ and $P_{para}$ versus $T$.

Magnetic Ellipsoid Shape Parameter ($T$)

The $T$ value varies between −0.37 to 0.69, but most the ellipsoids are plotted within the flattening field ($0 < T < 1$, planar fabric) (Figure 12c). The sites plotted within the prolate field ($T < 0$) are located mainly in high magnetic susceptibility sites (Figure 12a,c). This relation is also observable in the $K_m$ versus $T$ chart presented in Figure 12f, where the locations with the highest $K_m$ display prolate shapes. The spatial distribution of $T$ (Figure 12c) shows a symmetrical pattern, with an NNE–SSW axis of symmetry that coincides with the pluton long axis, similar to the $K_m$ spatial distribution.

### 5.2.2. Directional Magnetic Parameters

Based on the interpretation of the magnetic fabric, the orientation of the magnetic foliation is variable, ranging from NNW–SSE to NNE–SSW (Figure 13a$_1$). Generally, the magnetic foliations are sub-horizontal, featuring more vertical dips in the granite borders—mainly in the northeastern near the intersection of the ESE–WNW and the NNW–SSE fractures. The magnetic foliation arrangement displays concentric trajectories defining an elongated gently outward-dipping tongue shape with symmetry trending roughly NE–SW (Figure 13a$_2$). The orientation of the average magnetic foliation is N58° E; 5° SE (see the stereogram presented in Figure 13a$_1$), which in turn lies subparallel to the symmetry axis of the magnetic trajectories (Figure 13a$_1$,a$_2$). The scattering of magnetic lineations within the granite is characterized sub-horizontally along the NNE–SSW direction parallel to the granite's major axis; however, in the south–eastern border, the lineations tend to be parallel to the point of contact (Figure 13b$_1$). The magnetic lineation arrangement defines elongated trajectories (Figure 13b$_2$) with a symmetry axis concordant with the orientation of the average of magnetic lineation, which is 5°/N196° E (see the stereogram represented in Figure 13b$_1$).

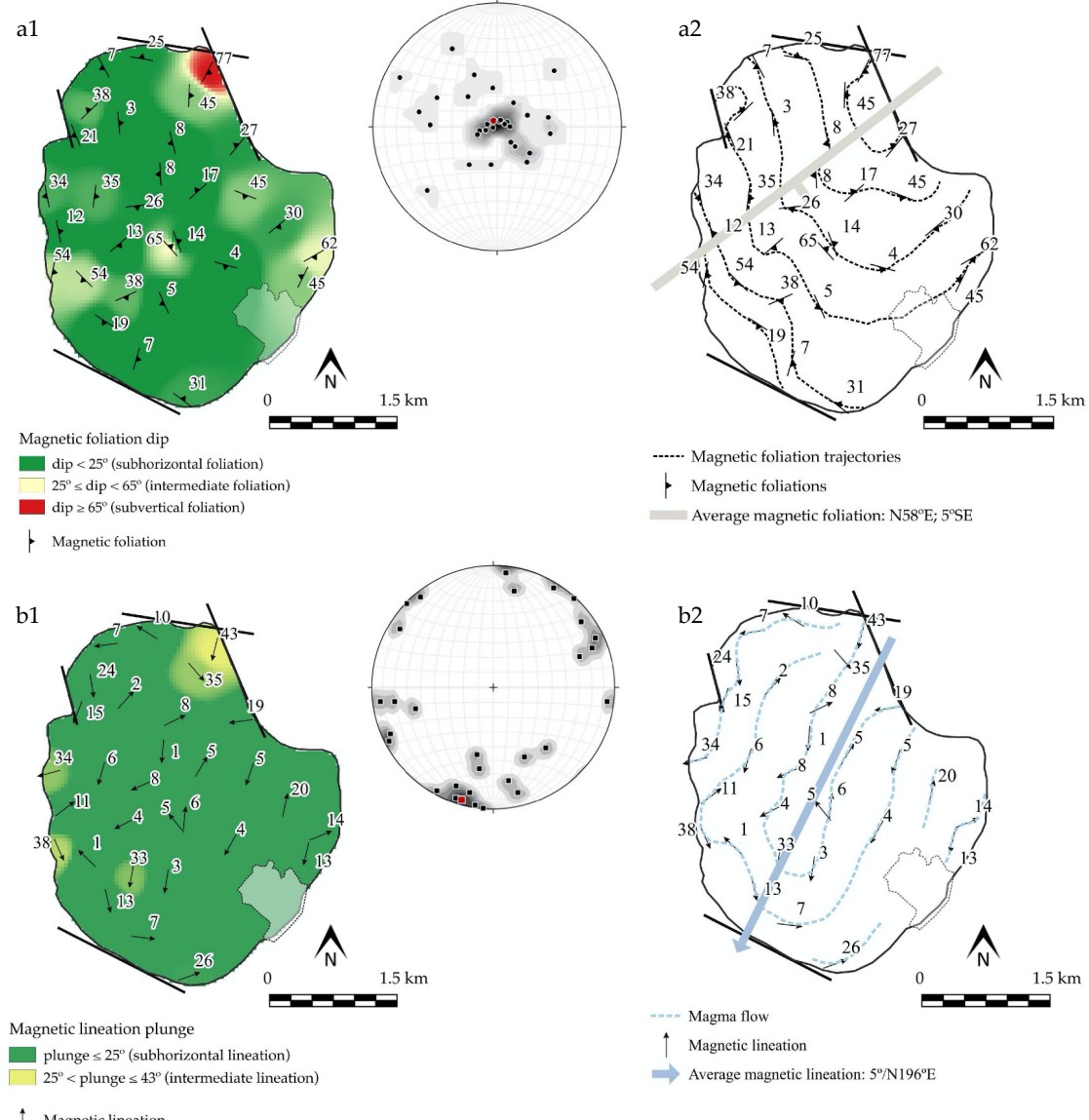

**Figure 13.** Magnetic foliation (perpendicular to $K_3$) and lineation ($K_1$) within the Capinha granite. Stereograms: equal-area, lower-hemisphere projection and contour intervals with uniform distribution; (**a1**) dip direction and dip of magnetic foliation at individual sampling sites. Stereogram of magnetic foliation poles. Inset map: variation in the dip angle of magnetic foliation; (**a2**) interpretative form–line map (magnetic trajectories) of magnetic foliations; (**b1**) plunge and azimuth of magnetic lineation at individual sampling sites. Stereogram of lineation orientations. Inset map: variation in the plunge angle of magnetic lineation; (**b2**) Interpretative magma flow based on magnetic lineations.

## 5.3. Fracturing

### 5.3.1. Outcrop Fracturing

The measurement of the outcrop fracturing in CG was made in situ in two distinct quarries using photo-interpreted faults, obtaining a total of 54 measurements. In general, the Capinha granite is moderated fractured. The main orientations obtained in the quarries are consistent with the photo-interpreted faults. The statistical study showed four representative fracture systems, 40° NE–50° NE (14.81%), 30° NE–N40° NE (11.11%), 20° NE–N30° NE (9.26%) and 0° NE–10° NE (11.11%), corresponding to 46.29% of the total measurements (Figure 14).

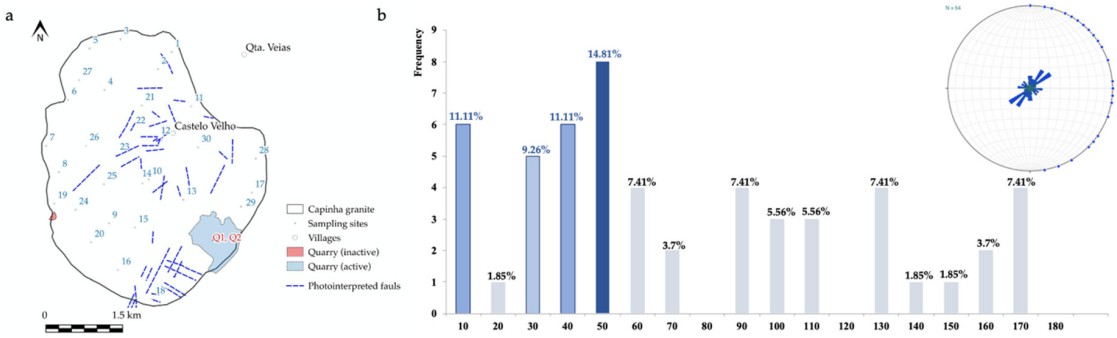

**Figure 14.** Outcropping scale fracturing in CGe. (**a**) Mapping of the photo-interpreted and regional fracturing systems visible in the CG; (**b**) histogram representing the distribution of the outcrop fracturing according to the 10° class subdivision and its corresponding rose diagram. Dark blue highlights the most important classes of the fracturing preferential orientations.

### 5.3.2. Microfracturing

The measurements of mode I microfractures in quartz (extensional or opening cracks, according to [116,117]) at a microscopic scale were used to better understand the evolution of regional stress. These cracks defined by "fluid inclusion planes" (FIP) were used as the markers of brittle deformation, as these sealed cracks do not affect mechanical continuity through the quartz crystal. The FIP appear aligned independent of the crystallographic orientation of the quartz crystals and cross-cutting quartz boundaries.

Microfracturing was studied in 8 thin sections and 118 microfractures were measured. The study and measurement of these microfractures showed that the granite did not exhibit strong brittle deformation at a microscopic scale. The statistical analysis showed a preferential orientation constrained into four systems: 40° N–50° E (16.24%), 50° N–60° E (11.11%), 60° N–70° E (10.26%) and 70° N–80° E (9.4%), representing 47.01% of the total measurements (Figure 15).

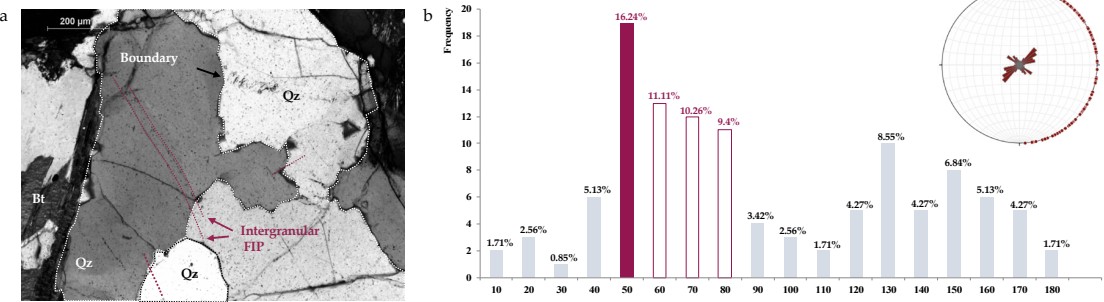

**Figure 15.** Statistical data obtained from the FIP measurements. (**a**) Microphotograph showing the general aspects of the intergranular FIP measured in quartz from the CG; (**b**) histogram representing the distribution of the FIP orientation according to the 10° class subdivisions and the corresponding rose diagram. Magenta indicates the most important classes of the FIP's preferential orientations.

## 6. Discussion

A combination of several approaches showed that the FSEC region is composed of a complex set of granitic facies intruding the metasedimentary sequences. The study on the magnetic parameters ($K_m$, $P_{para}$, $T$ and magnetic fabric) and geochemical signatures led to a more complete characterization of Capinha granite, helping us understand its spatial and genetic relationship with the host granites and evaluate its metallogenic potential.

### 6.1. Sequence of Magmatic Episodes

Capinha granite is peraluminous (molar A/CNK~1.35), muscovite-rich, relatively rich in potassium (ca. 4.4 wt%) and has high silica content (72.07 wt% to 73.41 wt%). PS and BC display similar characteristics but represent two independent episodes (PS is the older one). Fráguas granite is younger than BC granite and displays a metaluminous character (A/CNK ~0.96) (Figure 8b). Capinha and Fráguas represent independent and distinct magma pulses from PS and BC biotite-rich granites (Figures 9, 10b and 11).

The geochemical signatures of PS, BC, Fráguas and CG were compared to the melts produced by the experimental dehydration-melting of several distinct metasediments proposed in [118]. PS, BC and Fráguas could have formed by the partial melting of mafic pelites and greywackes [28] and Capinha granite (Tables 4 and 6) could have resulted from the partial melting of sources dominated by protoliths enriched in K feldspar [102,119,120], which is evidenced by their high-K calc–alkalic association (Figure 8a), low Ba and Sr and negative anomalies in $Eu_N$ (Figures 9 and 11).

The whole rock variation diagrams for major and trace elements (Figure 9) illustrate four distinct differentiation trends for the FSEC granites (PS, BC, Capinha and Fráguas), where the least evolved granite is PS and the most evolved is Fráguas, which is also supported by the Rb–Ba–Se ternary diagram (Figure 10b). The distinct and independent trends drawn by PS and BC are also supported by the REE diagrams, where the PS and BC displayed parallel REE profiles, with PS enriched in $REE^{(Tot)}$. The isotopic data obtained in [28] support the hypothesis of distinct source magmas for PS and BC, which present an εNd of −3.9 and −3.7 and a $(^{87}Sr/^{86}Sr)_{300}$ ratio of 0.7081 and 0.7076, respectively. The Fráguas and Capinha granites are neither related to each other nor to PS and BC, which is confirmed by the intersection of the REE patterns of Fráguas and Capinha in HREE. The Fráguas granite displays the highest negative anomaly, which suggests that this granite is more strongly enriched in plagioclase than the others. Although the Capinha REE pattern is parallel to the patterns of PS and BC REE, the $(Eu/Eu)_N$ negative anomaly is less marked, demonstrating a slight flattened REE profile, which is typical of granites emplaced in post-orogenic settings.

### 6.2. Interpretation of the Scalar Magnetic Parameters of Capinha Granite

Table 9 summarizes all the results obtained in the fieldwork, petrography, microfracturing and AMS approaches.

The $K_m$ in Capinha granite is, in general, homogeneous and does not display petrographic zonation. The abundance of muscovite compared to biotite is evident in the magnetic susceptibility, which is lower than 89.67 μS.I., a feature typical of muscovite > biotite ilmenite-type Portuguese Variscan granites ([121] and the references therein). The lowest values of $K_m$ (<30 μS.I.) are characterized by an enrichment of K feldspar crystals, which only occur on one site located in the western contact between the Capinha and BC granites.

The microstructural study in the CG displayed the predominance of magmatic to submagmatic microstructures corresponding to a $P_{para}$ of about 2.0%. However, a continuous transition was observed from magmatic to submagmatic microstructures in the core to low-T solid state microstructures in the peripheries. The low-T post-magmatic state is, essentially, characterized by a strong sericitization of the Na-plagioclase and by the occurrence of sutured borders and fracturing of quartz crystals (Figure 4 and Table 3). The highest $P_{para}$ values are only correlated to microdeformation since at the meso- and macroscale, the intrusion does not exhibit deformation. The evidence of low values for $P_{para}$ and the absence of deformation to the naked eye may indicate that the magnetic directional fabric observed in the Capinha granite was acquired during the magma ascent and emplacement.

The Fe-bearing silicate mineral (i.e., biotite) is responsible for the magnetic susceptibility magnitude and also for the ellipsoid magnetic shape (T). The spatial distribution of T in the granite displays an oblate shape, indicating a planar fabric, which is concordant with the lamellar shape of the biotite. However, linear fabrics are represented in the peripheries of the pluton. Considering that the present topography exposes the roof of the granitic suit based on sub-horizontal magnetic lineations and

foliations, a planar shape may be related to flattening against the overlying rocks before the erosion processes. The zones where the *T* shows prolate shapes and linear fabrics may indicate the granite feeder zone locations.

**Table 9.** Summary of the data obtained from the study of Capinha granite.

| Feature | Description |
|---|---|
| *Fieldwork* | |
| Altitude (min, max) | 461,670 m |
| Outcrop shape | Circular circumscribed body |
| Outcrop area | 7 km$^2$ |
| Relationship with surrounding rocks | Sharp contact between the granite and the surrounding rocks |
| Grain size | Medium-grained |
| Texture | Slightly porphyritic (K feldspar megacrysts ~1 cm) |
| Enclaves | No enclaves were observed |
| Fracturing | Moderately fractured: 40° N–50° E, 30° N–40° E, 20° N–30° E and 0° N–10° E, corresponding to 46.29%. |
| Deformation patterns | No deformation patterns were observed, except for the fracturing |
| *Petrography* | |
| Main mineralogy | Quartz (~40%), K feldspar (~25%) and Na plagioclase (albite and albite–oligoclase, ~20%) |
| Accessory mineralogy | Muscovite, biotite, chlorite, apatite, rutile, zircon, monazite and opaque mineralogy (~15%) |
| Opaque mineralogy | Ilmenite, hematite, arsenopyrite and pyrite |
| Later alterations | Sericitization of Na plagioclase, chloritization, the muscovitization of biotite and the oxidation of Fe-bearing minerals |
| Microstructures | Essentially magmatic to submagmatic with low-T solid stage microstructures |
| *Microfracturing* | |
| Fluid inclusion planes | 40° N–50° E, 50° N–60° E, 60° N–70° E, 70° N–80° E |
| Absolute frequency | 16.24%, 11.11%, 10.26% and 9.4%, respectively |
| *Anisotropy of magnetic susceptibility* | |
| Magnetic susceptibility ($K_m$) | 73.4 μS.I. |
| Paramagnetic anisotropy ($P_{para}$) | 2.0% |
| Magnetic ellipsoid shape ($T$) | 0.3 |
| Magnetic foliation | 58° NE; 5° SE |
| Magnetic lineation | 5°/196° N |

## 6.3. Magnetic/Magmatic Fabric in Capinha: Shape and Possible Feeder Zone

The evidence of consistently low values of $P_{para}$ in the whole Capinha granite and the absence of significative deformation patterns at distinct scales suggest that the spatial distribution and behavior of the magnetic fabric reflects the magmatic flow acquired during the ascent and emplacement of the Capinha magma.

The map of magnetic foliation trajectories (Figure 13a$_2$), obtained by simple extrapolation of the foliation traces at each site, clearly identifies an NE–SW elongated tongue-shape dome, whose core is located in the north–eastern boundary, where the dips are strong. Generally, in the western boundaries, the foliations display outward dips, which suggest continuity of the body under the PS granite. However, in the north–eastern boundary, foliations dip inward, corresponding to the close of the suite.

The orientation of the Fe-bearing minerals' major axis (mainly biotite), which corresponds to the magnetic lineations, may provide important information about the magma flow. In general, the lineations are sub-horizontal with NNE–SSW striking, which agrees with the major axes of the granite body. The spatial arrangement of the lineations describe an SSW magma flow (Figure 13b$_2$) with a presumed feeder zone in the north–eastern zone, where the lineation and foliation are vertical. The occurrence of overall sub-horizontal lineations suggests increases in horizontal movement that are higher than the vertical ones, indicating that the outcrop surface is the roof of the intrusion.

Considering these observations, we propose the presence of a possible feeder zone in the north–eastern boundary, where the foliations and lineations display vertical dips, $P_{\text{para}}$ exhibits higher values and the magnetic ellipsoid shape illustrates linear fabrics.

### 6.4. Integration of the Capinha Magnetic/Magmatic Fabric in Previous Studies

Previous studies performed in Serra da Estrela [122] that applied AMS and microstructural methods revealed that the emplacement of late kinematic granitoids was tectonically controlled in two main directions, 150° NE and 20° NE, which are well represented in the stretching of the magnetic lineations. The distribution and orientation of magnetic foliations found in this area were subdivided into two groups: (1) more or less circular intrusive plutons displaying steep concentric foliations (Serra da Estrela); and (2) in large granitic zones, namely on the eastern side of the Manteigas–Vilariça fault (MVF), where the foliations are medium to deep, trending NE–SW to N–S. The magnetic lineations displayed constant orientations over large areas (essentially NNW–SSE and NNE–SSW to N–S), with shallow to moderate plunges. In the eastern side of MVF, the lineations displayed preferential NNE–SSW to N–S orientations.

Geographically, Capinha granites are located on the eastern side of MVF (Figures 1 and 2). The magnetic fabric found for the Capinha is similar to that proposed by [122] for granites in the Covilhã–Guarda alignment; however, the dip in the magnetic fabric is shallower in Capinha compared to the others. This can be explained by the suit erosion level, where the outcrop of Capinha represents the roof of the body (see the previous explanation), and the granitic outcrops between Covilhã–Guarda (corresponding to biotite-rich granites) represent a deeper level.

### 6.5. The Fault System in the FSEC Region

The definition of the maximum compression stress ($\sigma_1$) orientation during late-Variscan fracturing is still a controversial subject of discussion. It is commonly considered that 312 Ma is the lower limit of the so-called Late-Variscan wrench-faulting period responsible for the fault system with a mean dextral strike of 25° NE and sinistral strike of N80° E. However, the 25° NE system was dextral in the late stages of the Variscan orogeny and sinistral during the Alpine Cycle, which affected all post-D$_3$ granites (e.g., [57]). In north and central Portugal, four main sub-vertical fault systems were recognized: 25° N ± 25° E (or NNE–SSW), 80° N ± 20° E (ENE–WSW), 155° N ± 15° E (or NW–SE) and 110° N ± 10° E (or WNW–ESE). The 155° N ± 15° E and the 110° N ± 10° E systems display a complex history inherited from the early Variscan deformation (e.g., [44,54,57]).

The fracturing at outcrop scale in the Capinha granite is homogeneous, ranging between N20° and 50°E (Figure 14), fitting in the 25° N ± 25°E (or NNE–SSW) strike–slip system. On the other hand, the microfracturing represented here by the fluid inclusion planes exhibits heterogeneous fracturing at 25° N ± 25° E and 80° N ± 20° E. Thus, it is suggested that during the ascent, emplacement, and complete consolidation of Capinha granite, the main compression stress was NE–SW oriented, which is demonstrated by the magnetic directional data and in the first brittle deformation marked in the FIP orientation. Later, the maximum compression rotated NNW–SSE. During the Alpine orogeny, a second stage of brittle deformation developed NE–SW (20° N –50° E) fractures in the granite.

### 6.6. Tectonic Constraints and Emplacement Mechanism

The Capinha granite intruded into the PS late-Variscan granite, dated ca. 304.1 ± 3.9 Ma and classified as late-D$_3$ [83] (Figures 2 and 3).

The importance of the N150°–155° E and N20°–25° E strike–slip faults in the Serra da Estrela region were previously mentioned [122] in the ascent and emplacement of the late- to post-D$_3$ biotite-rich granites. The reactivation of these sets of strike–slip faults is well recorded in the region (see Section 3. *Fundão–Serra da Estrela–Capinha region*). The strong reactivation of the N25° ± 25° E strike–slip faults, inherited from the D$_3$-Variscan phase and parallel to the MVF, provided zones for the emplacement of late-Variscan granites, which, in turn, were cut by quartz veins in the same direction (Figures 2 and 3).

The N155° ± 15° E strike–slip faults are observable in the southern contact between the PS and BC granites.

In Capinha granite, the N155° ± 15° E system is poorly represented and only marked by some lineations in the north zone (and only in ca. 5% of the regional fracturing (Figure 14)). On the other hand, the N25° ± 25° E system is recorded by magnetic lineations in the whole granite structure (Figure 13b) and also in brittle deformation at several scales. Therefore, we propose that the ascent and emplacement of Capinha magma was controlled by the intersection between the dextral N25° ± 25° E and the dextral N155° ± 15° E strike–slip faults, where the latter is less expressive and acted only as channel conduit for the ascent of the magma. The N25° ± 25° E fault strongly controlled the orientation of the magnetic lineations and, consequently, the magma flow. The intersection between these two systems is marked in the northern boundary of the Capinha.

The emplacement of granites is still a frequent topic of discussion, particularly concerning the mechanisms of pluton construction ([20,123–129] and the references therein). The Capinha granite clearly cut off contact between PS biotite-rich granites and metasedimentary sequences and its emplacement does not suggest deformation in the host rocks, indicating passive emplacement. The evidence of intense fracturing at a regional scale may facilitate the ascent and emplacement of the magma [130–132]. In this way, the magma ascent was controlled only by tectonics related to the strike–slip system, which helped the magma rise due to the buoyancy forces resulting from the density difference between the magma (usually less dense) and the country rock. The lateral spread of the magma occurred in the interface between lithologies with contrasting rheologic behaviors. The progression stopped due to an increase in magma viscosity, and when the magma approached a solidus state, it acquired high $P_{para}$ in the pluton peripheries and low-T solid state microstructures.

### 6.7. FSEC Granites and Their Metallogenic Implications

Sn (Li) mineralizations occur mainly in aplite–pegmatite sills and quartz veins that are generally intrusive in biotite-rich granites. The north–east area of Belmonte-Gonçalo, contains aplite–pegmatite sills with lepidolite and cassiterite (Li–Sn) intrusive in late-$D_3$ biotite-rich coarse-grained porphyritic granites (Peroviseu–Seia), as well as in the exocontact of late- to post-$D_3$ two micas granite (e.g., Fráguas granite). This spatial association suggests that Sn (Li) mineralizations are associated with Fráguas granite (Sn~62 ppm) [70]. This granite's relationship to the mineralizations, geochemical signatures and evolved degree (Figure 10) of the Fráguas granite allows us to classify it as an Sn-bearing granite [133].

W and W–Sn mineralizations occur in the quartz veins hosted in the PS and Capinha granites and cut the geological contact between the PS and SGC. Several W and W–Sn occurrences were recognized in the Capinha area (Figure 16):

1.  W quartz veins in the contact SGC/Peroviseu–Seia granite near Capinha (e.g., Peroviseu, Mata, Ribeiro do Salgueiro, Caverna and Covões);
2.  W quartz veins in Capinha granite (e.g., Castelo Velho);
3.  W (Sn) quartz veins in the Peroviseu–Seia granite near Salgueiro (e.g., Pombal Lameiros, Veias and Malta).

The preferential occurrence of several W and W–Sn mineralizations near the Capinha granite suggests that the mineralization is related to the emplacement of this post-$D_3$ granite, which is enriched in Sn (~17 ppm) and W (~10 ppm).

As mentioned previously, similar W–Sn mineralizations occur in the SGC where no granites outcrop, with Panasqueira being an example of this kind of ore deposit (and where a two-mica granite was recognized in drill cores). The geological settings of the W and W–Sn mineralizations in Capinha and Panasqueira are similar. The emplacement of post-tectonic Capinha granite and the greisenized cupula linked to the two-mica granite in Pasqueira suggest that the emplacement of post-$D_3$ peraluminous granites played an important role as a source of W–Sn metals and heat, which stimulated the remobilization of metal enriched fluids and the resulting concentration of spatially associated W–Sn

ore deposits. Primary W and W–Sn deposits are the result of hydrothermal systems commonly related to magmatic rocks of a granitic composition and are developed in apical high-level portions of granites, e.g., Capinha, where W accumulated in residual evolved fluids during fractional crystallization.

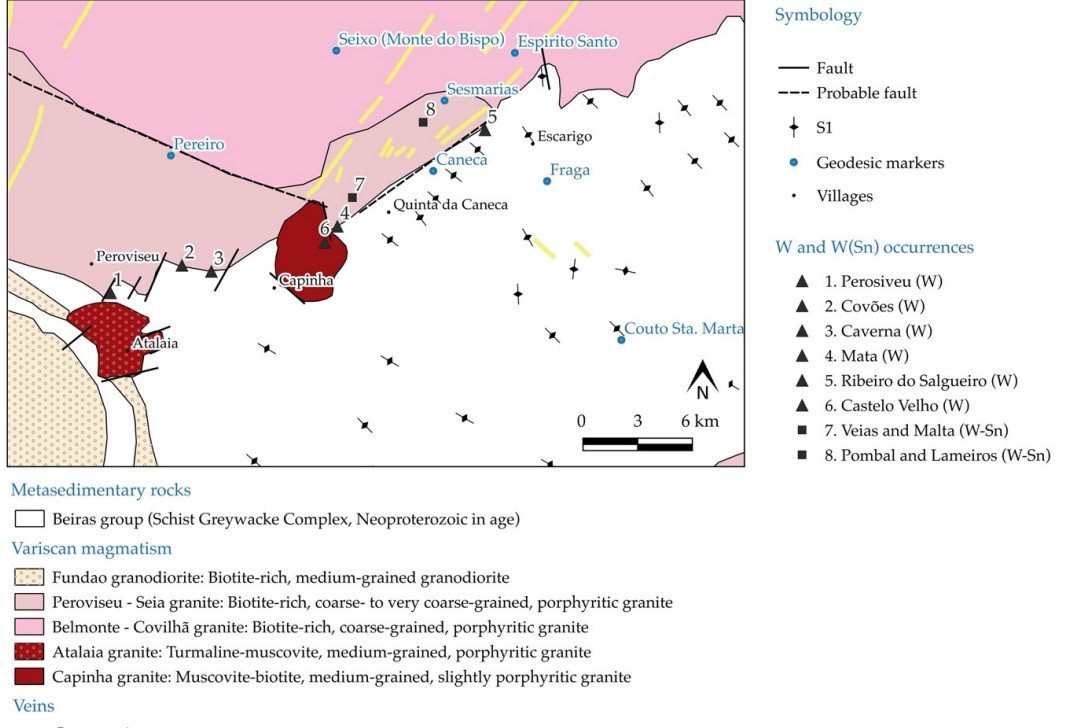

**Figure 16.** Geological sketch map of the Capinha area showing the association between the outcrop lithologies and the W and W–Sn metallic occurrences [66,67,81,82]. Note: the precise localizations of the metallic occurrences were taken from the *Sistema de Informação de ocorrências e Recursos Minerais Portugueses* (SIORMP) [134].

### 6.8. Capinha Granite: Syn- and Post-Emplacement Model

Based on the previous data and subsequent discussion, a four-stage model of emplacement for the post-D$_3$ Capinha granite (Figure 17) is proposed:

**Stage I:** In the late-Variscan extensional setting, the reactivation of 25° N ± 25° E and 155° N ± 15° E strike–slip crustal scale faults led to the opening of spaces, which acted as feeder channels for the passive ascent of the Capinha magma. The feeder zone was located in the northern boundary, as evidenced by the high values of $P_{para}$, prolate ellipsoid shapes and the vertical magnetic fabric.

**Stage II:** The vertical ascent of magma through the dykes along the crustal strike–slip faults was facilitated by buoyancy forces. The existence of a contrasting rheologic boundary favors lateral magma spread from NNE to SSW, forming an asymmetric tongue-shaped laccolith.

**Stage III:** During the ascent, the magma underwent magmatic differentiation processes, with the residual magmatic–hydrothermal fluid becoming enriched in incompatible (Rb and W) and volatile elements. This fluid's geochemical signature and spatial relationship with the W and W (Sn) mineralizations suggest that Capinha played an important role in the metallic concentrations.

**Stage IV:** After complete crystallization, the Capinha granite experienced its first brittle deformation event, which is recorded in the NE–SW to ENE–WSW FIP and is consistent with NE–SW oriented σ1. Later, the rotation of the maximum compression from NE–SW to NNW–SSE led to a second stage of brittle deformation, which generated NE–SW fractures in the granite.

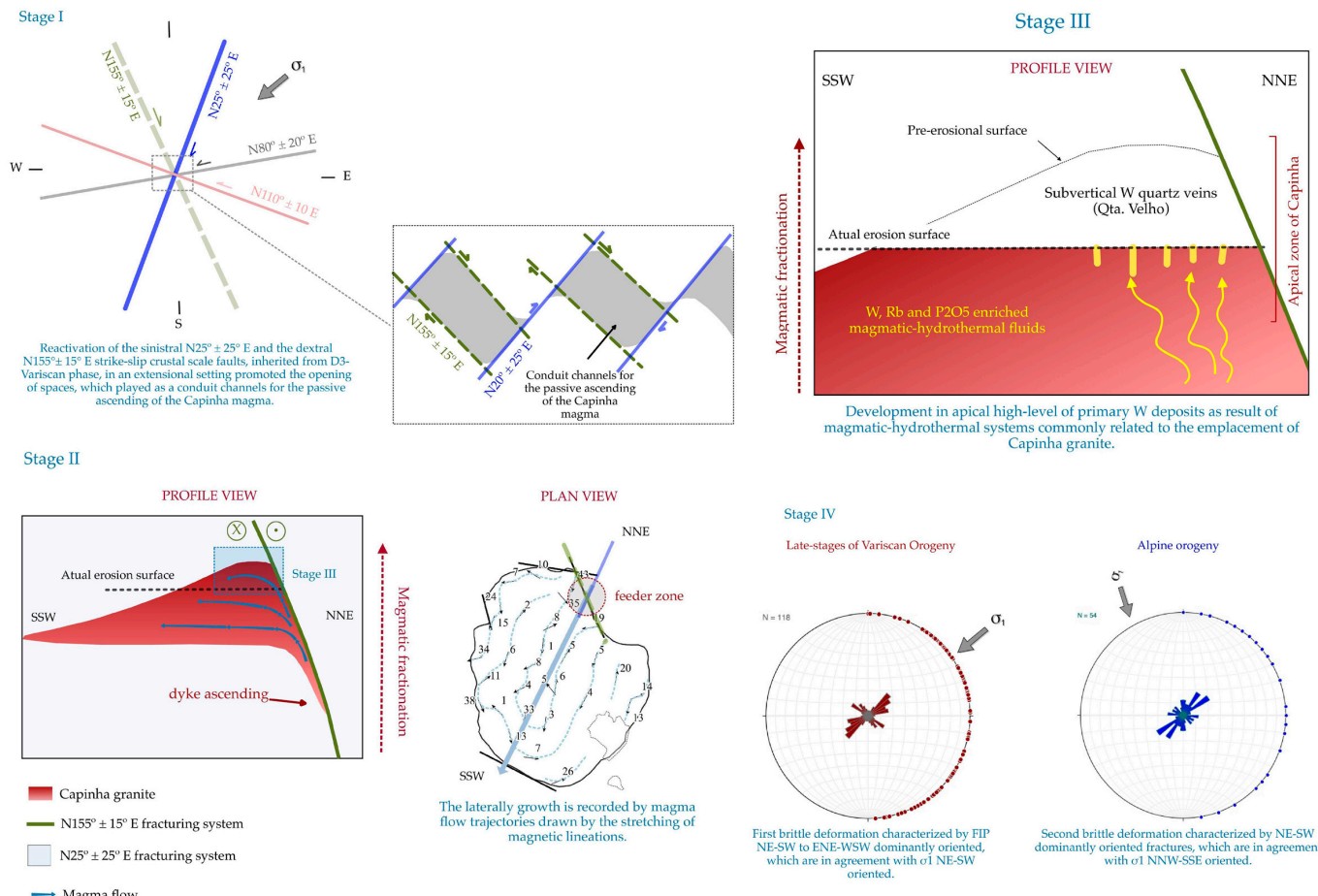

**Figure 17.** Schematic model proposed for the ascent, emplacement and brittle deformation of the Capinha granite.

## 7. Conclusions

The geochemical data together with the microstructures and the AMS results provided the following conclusions:

- Studying the geochemical behaviors of FSEC granites allowed us to identify distinct and independent differentiation trends, namely, Peroviseu–Seia (PS), Belmonte–Covilhã (BC), Fráguas and Capinha. The Fráguas and Capinha are more evolved than BC and BC.
- Capinha granite has a homogeneous spatial distribution in $K_m$, displaying values below 73.4 µSI, thereby indicating that magnetic mineralogy is dominated by paramagnetic minerals such as biotite, which is classified as an ilmenite-type granite. The generally low magnetic anisotropy in the whole area is typical of post-Variscan muscovite–biotite granites, where $P_{para}$ corresponds to a weak orientation of the fabric and also a low degree of deformation. The highest values were found in the peripheries, especially in the north–eastern direction, corresponding to the late-deformation patterns caused by the solidification of magma against the host rocks. The magnetic ellipsoid shapes are, in general, oblate, indicating a planar fabric, which may indicate the roof of the granite suit. Prolate shapes were found in the north–eastern zone, fitting the zones with high $P_{para}$.
- The granite displays NE–SW striking foliations associated with NNE–SSW magnetic lineations parallel to the long axis of the pluton, showing that the acquisition of the magnetic fabric resulted from magma stretching parallel to the magma flow (from NNE to SSW), at the end of the last ductile Variscan deformation phase ($D_3$). This enhanced the role of 155° N ± 15° E and 25° N ± 25° E structures on the magma's ascent and emplacement. The general sub-horizontal magnetic fabric suggests that the outcrop is the roof of the suit. Vertical foliations associated with vertical lineations in the north–eastern boundary suggest the location of a deep feeder zone.
- The ascent of the magma in the dyke structures at high structural levels, as well as the lateral spread of the magma in the contrasting rheologic boundary, allowed the construction of an asymmetric tongue-shaped laccolith, with its root located in the north–eastern zone.
- After complete consolidation, the late-Variscan faulting indicates the reactivation of inherited structures, and the first brittle deformation was recorded in the FIP, with the main compression being oriented NE–SW. After the rotation of the maximum compression to NNW–SSE, during Alpine orogeny, a second stage of brittle deformation was registered, developing fractures in the granite (generally NE–SW (20° N–50° E) oriented).
- The FSEC region is characterized by several metallic occurrences, and the spatial distribution of the distinct types of mineralizations is not accidental. The Sn mineralizations in pegmatites and quartz veins with cassiterite are hosted by PS and BC granites, but only those near Fráguas granite, suggesting that the mineralization source is related to the Fráguas (Sn ~62 ppm) granite. Therefore, the primary W and W–Sn deposits are the result of hydrothermal systems commonly related to magmatic rocks with a granitic composition and are developed in apical high-level portions of granites, such as Capinha and Panasqueira.

**Author Contributions:** Conceptualization, A.G., H.S. and F.N.; data curation, A.G., H.S. and F.N.; formal analysis, A.G., H.S. and F.N.; funding acquisition, H.S. and F.N.; investigation, A.G., H.S. and F.N.; methodology, A.G., H.S. and F.N.; project administration, A.G. and H.S.; resources, A.G., H.S. and F.N.; software, A.G., H.S. and F.N.; supervision, H.S. and F.N.; validation, A.G., H.S. and F.N.; visualization, A.G., H.S. and F.N.; writing—original draft, A.G., H.S. and F.N.; writing—review and editing, A.G., H.S. and F.N. All authors have read and agreed to the published version of the manuscript

**Funding:** This research was funded by the Fundação para a Ciência e Tecnologia (FCT), Grant Number SFRH/BD/115324/2016.

**Acknowledgments:** This work was supported by the Portuguese Foundation for Science and Technology (FCT) project UIDB/04,683/2020—ICT (Institute of Earth Sciences). The authors thank the POCTEP—Interreg Project 0284_ESMIMET_3_E "Development of exploitation environmental and energy techniques in metallic mining" for their technical support. The authors thank the Department of Geosciences, Environment and Spatial Planning at the Faculty of Sciences of the University of Porto for making their laboratories available for carrying out the

studies presented in this study. The authors thank Duarte Silva and José Carlos Oliveira for their help in the sampling and fieldwork and Cândida Neto for her help in the SEM-EDS analysis. The authors deeply thank Lourenço Pereira from the CIMPOR, AGREPOR, SA, Centro de Exploração do Fundão, for the samples collected in the quarry for whole rock geochemistry. The authors thank the anonymous reviewers for their helpful comments and suggestions, which clearly improved the quality of the manuscript.

**Conflicts of Interest:** The authors declare no conflicts of interest. The funders had no role in the design of the study; in the collection, analyses or interpretation of the data; in the writing of the manuscript or in the decision to publish the results.

## Appendix A

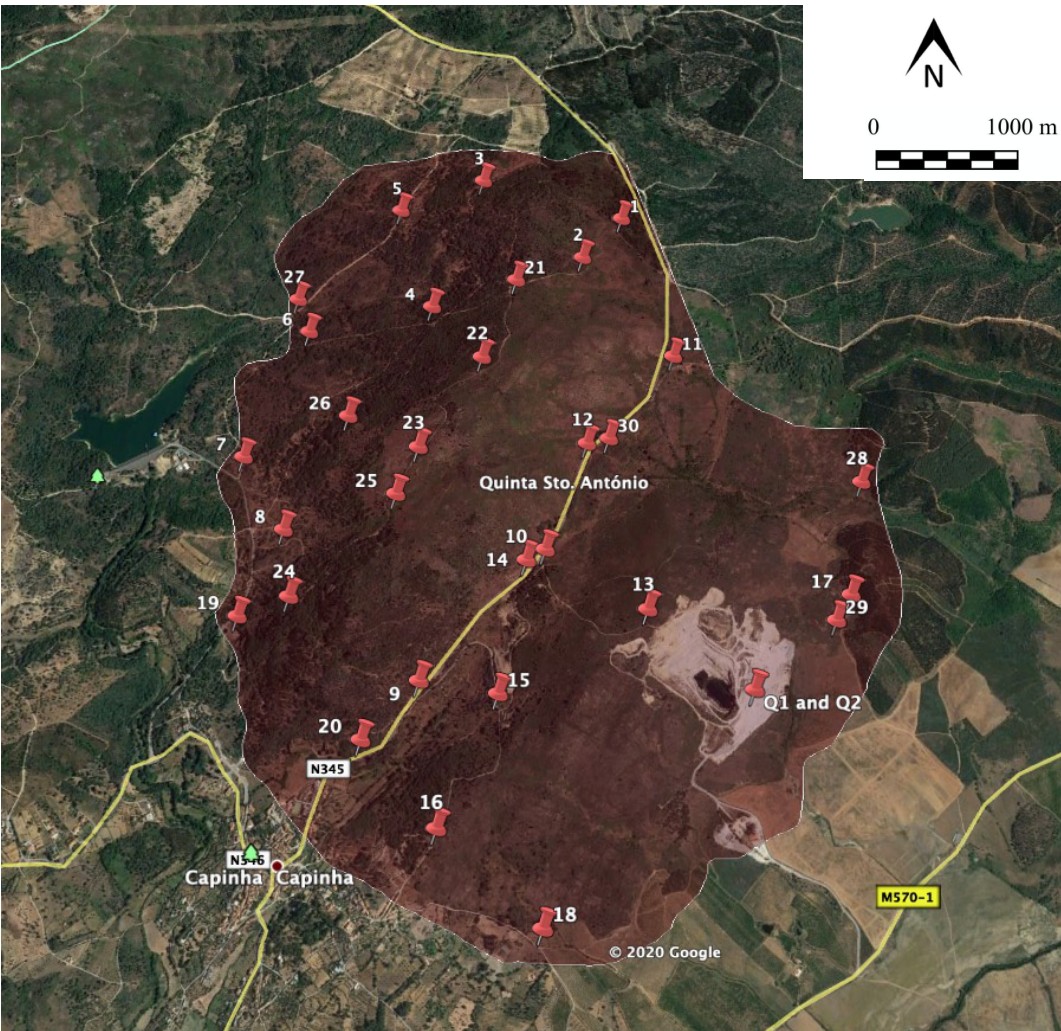

**Figure A1.** Location of the sampling sites for the Capinha granite (Google Earth image accessed on 5 May 2020).

**Table A1.** Geographic coordinates, elevations, methods applied, and number of samples collected from each sampling site in Capinha.

| ID | Latitude | Longitude | Elevation (Above Sea) | Methods Applied | Samples Collected |
|---|---|---|---|---|---|
| 1 | 40°13′23.520″ N | 7°21′41.760″ W | 585 m | Petrography, microstructures, whole rock geochemistry, anisotropy of magnetic susceptibility and fluid inclusion planes | 4 oriented cores |
| 2 | 40°13′17.400″ N | 7°21′48.240″ W | 600 m | Anisotropy of magnetic susceptibility | 6 oriented cores |
| 3 | 40°13′28.200″ N | 7°22′5.520″ W | 570 m | Anisotropy of magnetic susceptibility | 6 oriented cores |
| 4 | 40°13′10.200″ N | 7°22′13.080″ W | 598 m | Petrography, microstructures, whole rock geochemistry, anisotropy of magnetic susceptibility and fluid inclusion planes | 5 oriented cores |
| 5 | 40°13′24.960″ N | 7°22′19.920″ W | 577 m | Anisotropy of magnetic susceptibility | 5 oriented cores |
| 6 | 40°13′7.680″ N | 7°22′33.960″ W | 541 m | Anisotropy of magnetic susceptibility | 4 oriented cores |
| 7 | 40°12′50.639″ N | 7°22′42.637″ W | 518 m | Anisotropy of magnetic susceptibility | 7 oriented cores |
| 8 | 40°12′41.040″ N | 7°22′34.680″ W | 529 m | Anisotropy of magnetic susceptibility | 4 oriented cores |
| 9 | 40°12′22.680″ N | 7°22′11.640″ W | 519 m | Anisotropy of magnetic susceptibility | 5 oriented cores |
| 10 | 40°12′38.160″ N | 7°21′52.920″ W | 564 m | Petrography, microstructures, anisotropy of magnetic susceptibility and fluid inclusion planes | 5 oriented cores |
| 11 | 40°13′4.071″ N | 7°21′32.912″ W | 546 m | Petrography, microstructures, whole rock geochemistry, anisotropy of magnetic susceptibility and fluid inclusion planes | 5 oriented cores |
| 12 | 40°12′52.001″ N | 7°21′46.878″ W | 556 m | Anisotropy of magnetic susceptibility | 5 oriented cores |
| 13 | 40°12′30.841″ N | 7°21′36.885″ W | 568 m | Anisotropy of magnetic susceptibility | 4 oriented cores |
| 14 | 40°12′36.830″ N | 7°21′55.803″ W | 567 m | Anisotropy of magnetic susceptibility | 4 oriented cores |
| 15 | 40°12′21.098″ N | 7°21′59.448″ W | 533 m | Petrography, microstructures, whole rock geochemistry, anisotropy of magnetic susceptibility and fluid inclusion planes | 5 oriented cores |
| 16 | 40°12′6.106″ N | 7°22′7.633″ W | 516 m | Anisotropy of magnetic susceptibility | 5 oriented cores |
| 17 | 40°12′33.197″ N | 7°21′5.113″ W | 497 m | Petrography, microstructures, anisotropy of magnetic susceptibility and fluid inclusion planes | 5 oriented cores |
| 18 | 40°11′55.571″ N | 7°21′51.777″ W | 474 m | Petrography, microstructures, whole rock geochemistry, anisotropy of magnetic susceptibility and fluid inclusion planes | 5 oriented cores |
| 19 | 40°12′30.487″ N | 7°22′40.665″ W | 506 m | Petrography, microstructures, whole rock geochemistry, anisotropy of magnetic susceptibility and fluid inclusion planes | 5 oriented cores |
| 20 | 40°12′15.987″ N | 7°22′19.970″ W | 499 m | Anisotropy of magnetic susceptibility | 5 oriented cores |
| 21 | 40°13′12.7452″ N | 7°21′42.768″ W | 586 m | Anisotropy of magnetic susceptibility | 5 oriented cores |
| 22 | 40°13′1.92″ N | 7°22′3.774″ W | 673 m | Anisotropy of magnetic susceptibility | 5 oriented cores |
| 23 | 40°12′50.1732″ N | 7°22′13.098″ W | 655 m | Anisotropy of magnetic susceptibility | 4 oriented cores |
| 24 | 40°12′32.3496″ N | 7°22′32.0952″ W | 571 m | Anisotropy of magnetic susceptibility | 5 oriented cores |
| 25 | 40°12′44.514″ N | 7°22′16.3416″ W | 626 m | Anisotropy of magnetic susceptibility | 5 oriented cores |
| 26 | 40°12′54.9864″ N | 7°22′25.2372″ W | 606 m | Anisotropy of magnetic susceptibility | 5 oriented cores |
| 27 | 40°13′9.9372″ N | 7°22′38.0388″ W | 523 m | AMS Anisotropy of magnetic susceptibility | 5 oriented cores |
| 28 | 40°12′47.7612″ N | 7°21′2.3796″ W | 482 m | Anisotropy of magnetic susceptibility | 5 oriented cores |
| 29 | 40°12′36.234″ N | 7°21′8.01″ W | 500 m | Anisotropy of magnetic susceptibility | 6 oriented cores |
| 30 | 40°12′52.7947″ N | 7°21′43.3656″ W | 545 m | Anisotropy of magnetic susceptibility | 6oriented cores |
| Q1 and Q2 | 40°12′21.91″ N | 7°21′20.18″ W | 481 m | Whole rock geochemistry | About 20 kg |

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
