# Peer review of "Geochemical Signature and Magnetic Fabric of Capinha Massif (Fundão, Central Portugal): Genesis, Emplacement and Relation with W–Sn Mineralizations"

_minerals, doi:10.3390/min10060557_

Round 1

Reviewer 1 Report

The paper "Geochemical signature and magnetic fabric of Capinha massif (Fundão, Central Portugal): genesis, emplacement and relation with the W-Sn mineralizations" submitted by Ana Gonçalves and co-workers presents field and petrographic relationships, including a microstructural analysis, whole rock compositions and petrophysical data (anisotropy of magnetic susceptibility) of the Capinha granite from the Fundão-Serra da Estrela-Capinha (FSEC) region. The work also discusses geochemical relationships of this granite with other plutons from the area and analyzes its spatial and genetic relationships with the host rocks, proposing a mechanism of magma emplacement.

The work is interesting and suitable for publication in Minerals. However, English style is confusing, needing some sentences rephrasing, and there are too many typos (use of verb tenses, subject-verb agreement, and use of adverbs and adjectives) that must be corrected before accptance.

The following papers are relevant for the geochronology of the various magmatic and tectonic events discussed in the work and should be cited :

Bea, F., Pesquera, A., Montero, P., Torres-Ruiz, J., Gil-Crespo, P.P., 2009. Tourmaline 40Ar/ 39Ar chronology of tourmaline-rich rocks from Central Iberia dates the main Variscan deformation phases. Geologica Acta 7, 399–412.

Scarrow,J.H., Molina,J.F.,Bea,F., Montero, P.,Vaughan,A.P.M.,2011.Lamprophyre dikes as tectonic markers of late orogenic transtension timing and kinematics: a case study from the Central Iberian Zone. Tectonics 30, TC4007. doi:10.1029/2010TC002755.

I consider that the paper can be accepted after minor revisions.
In the annotated pdf I provide detailed comments and corrections.

Best regards.

Author Response

Sincerely,

Ana Gonçalves

Reviewer 2 Report

This article describes the structural and chemical characteristics of the Capinha massif. This article is particularly of interest because of its location (one of the richest in W in Europe) and its approach. While I am not a specialist of structural geology, I appreciate the effort of the author to provide this study. This is not a common approach in the "W world" and it is needed. On the geochemical aspect, there is nothing breakthrough but it is done correctly and the readers might appreciate the regional data so I have nothing major to say.

As said, I am not a structural geologist so I didn't have any comments on that part of the manuscript.

Overall, it is a good manuscript. There is attention to detail, the descriptions are thorough and the figures of good quality (just be careful to export them in high quality). However, I would work a little bit on the English. It is totally understandable but some phrasing sound strange and it might discredit the manuscript for the wrong reasons.

The few comments I had, are in the file attached.

Author Response

Sincerely,

Ana Gonçalves

Reviewer 3 Report

The paper is very interesting and flawless at a first glance, but after careful reading it seems that it has some drawbacks. The problem is that the pieces of information given are not homogeneous for every one magmatic item of the whole sequence of plutons. Generally the joint information about geochemical signature / petrography and magnetic fabric is related to Capinha pluton only. The main target shown is to reconstruct the mode of emplacement and to give a characteristics of the granitic melt. So what for the Authors need the detailed description of geochemistry and petrography of the whole Variscan granite series. Even within granite series the very early pre-Variscan is mentioned and after it is not considered in further research. All others (Variscan) have detailed description (petrography, some geochemical data), but no one information about their magnetic fabric is given. For Authors the emplacement process is interested in term of Capinha research only. The data and discussion on them are excellent. But the data are only for Capinha and not for any other pluton. In this sense, the construction of the paper is defective. Some research / insights are developed in excess for all plutons, other data are selective and fractional given only for one of them.

But the paper and the data given are very interesting. I would supposed to select all data – geochemistry, petrography and magnetic fabric for Capinha only. Much shorter paper and much more comprehensive. The paper has in title Capinha and suggested is a research on Capinha. Comparative study would be good if you would have a set of magnetic fabric data for all plutons. It is not the case. The part devoted to the issues recognizing mode of magma emplacement is significant, hence the impression of non-homogeneity of the entire article is clear, because of lack of similar information about other plutons. I would define it as a “shock”, like a clear break in the logic of work.

Also the geochemical data presented for all plutons are not entirely convincing. Certainly the magmatic bodies are evolved by different magma evolution mechanisms, but the amount of data is scarce. Even K/Rb as a differentiation factor is not convincing without detailed recognition how K and Rb are incorporated into minerals during magma evolution. For instance Kd for K-feldspar can be >1 or <1, e.g. compatible or incompatible. I think the paper would benefit if these parts (comparative study, which requires refinement) were removed. The paper is elegant without them. Conclusions drawn on basis on the comparative study are poor and you may draw them without any sophisticated discussion. Enough to look at A/NK vs A/CNK diagram.

BTW I don’t believe you are able to recognize orthoclase under polarizing microscope. It is a matter of order-disorder in the feldspar structure, so the right tool is XRD.

Recommendation - minor revision.

Author Response

Sincerely,

Ana Gonçalves
